# Revealing the intrinsic nature of the mid-gap defects in amorphous Ge$_2$Sb$_2$Te$_5$

Konstantinos Konstantinou [1], Felix C. Mocanu [1], Tae-Hoon Lee[1] & Stephen R. Elliott[1]

Understanding the relation between the time-dependent resistance drift in the amorphous state of phase-change materials and the localised states in the band gap of the glass is crucial for the development of memory devices with increased storage density. Here a machine-learned interatomic potential is utilised to generate an ensemble of glass models of the prototypical phase-change alloy, Ge$_2$Sb$_2$Te$_5$, to obtain reliable statistics. Hybrid density-functional theory is used to identify and characterise the geometric and electronic structures of the mid-gap states. 5-coordinated Ge atoms are the local defective bonding environments mainly responsible for these electronic states. The structural motif for the localisation of the mid-gap states is a crystalline-like atomic environment within the amorphous network. An extra electron is trapped spontaneously by these mid-gap states, creating deep traps in the band gap. The results provide significant insights that can help to rationalise the design of multi-level-storage memory devices.

[1] Department of Chemistry, University of Cambridge, Lensfield Road, Cambridge CB2 1EW, UK. Correspondence and requests for materials should be addressed to K.K. (email: kk614@cam.ac.uk)

Phase-change materials based on chalcogenide alloys, such as Ge–Sb–Te, display fast and reversible transformations between a conductive crystalline structure and an amorphous phase that exhibits a higher electrical resistivity[1,2]. The two states show very different electronic-transport and optical properties, leading to a unique set of features, exploited in non-volatile phase-change random-access electronic memory (PCRAM), which is a candidate of digital-storage technology to replace the current Si-based flash memory[3].

One of the technological challenges to make PCRAM competitive with flash is to increase their storage density. Multi-level-storage operation would enable not only a manufacturing cost reduction but also the desired enhancement of the storage density[4]. Multi-level-storage stores information as multiple intermediate resistance levels in between the high-resistance, entirely amorphous phase and the low-resistance, entirely crystalline state[5], thereby enabling the storage of multiple bits per cell[6].

However, the development of multi-level storage PCRAM devices has been hindered owing to the time-dependent, resistance-drift phenomenon in the amorphous phase. This has been related to the instability of the amorphous state of the phase-change material, resulting in so-called material ageing, characterised by a logarithmic increase of the resistivity with time[7]. This increase of the cell resistance therefore negatively impacts on data storage in multiple intermediate memory levels and can lead to memory failure[4].

The lack of long-range order in amorphous semiconductors can give rise to the formation of defect-related electronic states within the band gap, which correspond to an intrinsic property of the disordered material[8]. Resistance drift and threshold-voltage switching have been attributed to localised states in the band gap[9–12], and also to the electron-trapping kinetics associated with these defect-related electronic states[13–16]. Te–Te chains involving coordination defects have been considered as the defect states present in phase-change materials, as in the Valence-Alternation Pair model[17,18]. However, the presence of Te–Te chains in amorphous $Ge_2Sb_2Te_5$ has been questioned experimentally[19] and from density-functional-theory (DFT) calculations[20,21], as the average nearest-neighbour Te–Te coordination number in the glass structure has been measured to be very small. In addition, Caravati et al.[22] identified a mid-gap defect state with Sb–Te chains in an amorphous $Ge_2Sb_2Te_5$ model, generated by ab initio molecular dynamics simulations. However, their mid-gap defect state is rather delocalised through the entire simulation cell rather than being localised at a specific structural motif.

The most significant complication for modelling defects in amorphous materials concerns variations in their local environment caused by structural disorder. Any simulation study should involve a statistical analysis of many different models to investigate the probability of defect formation and to obtain distributions of their properties. The gap-state density of glassy $Ge_2Sb_2Te_5$ has been measured by modulated photocurrent experiments[23] and was found to be $1 \times 10^{10}$/cm$^3$/eV for mid-gap defect states and $4 \times 10^{10}$/cm$^3$/eV for shallow gap states (located at ~0.2 eV from the valence-band edge). Thus, the concentration of these electronic states is relatively low, making these defects difficult to be studied by computational modelling. Hence, calculations for many models are required to credibly predict the mid-gap defect-state configurations, and to identify the atomic environment that can host these special electronic states within the glass network (see Supplementary Note 1 for further details).

In this study, we take advantage of a machine-learned interatomic potential, recently developed by our group, for amorphous $Ge_2Sb_2Te_5$[24], in order to generate an ensemble of uncorrelated 315-atom glass models of this material. Subsequently, DFT calculations, using non-local functionals, are employed to optimise the geometries of the generated glass models and to calculate their electronic structure. The non-local functional is able to provide a very good estimate of the band gap and it can identify the atomic structural features responsible for the mid-gap localised defects. These calculations are extended to a 900-atom amorphous model to enhance the quality of the observations on the atomistic nature of the mid-gap defect states expected to be present in a real sample of glassy $Ge_2Sb_2Te_5$. Finally, we explore the capability of the mid-gap defect states to serve as electron traps, providing further insight into the performance of these PCRAM devices.

## Results and discussion

**Glass validation.** The present calculations make use of a machine-learned interatomic potential and DFT. In order to create sufficient statistics, 30 independent 315-atom models of amorphous $Ge_2Sb_2Te_5$ (225GST) were generated using classical molecular dynamics (MD) simulations with a Gaussian approximation potential (GAP), followed by periodic hybrid-DFT calculations of the electronic structure of these models, as described in the Methods section. It is noted that GAP-MD simulations have been successfully used recently to generate realistic and accurate glass models of other compounds as well[25].

The GAP potential exhibits an accuracy close to that of the underlying DFT-PBEsol training set[24]. In addition, we have demonstrated that the potential can be employed for rigorous modelling, with near-DFT accuracy, of the short- and medium-range-order structure of amorphous 225GST. Analysis of the local atomic structure showed that, even though the GAP potential may slightly under-estimate the proportion of tetrahedral Ge environments and the amount of homopolar bonds in the glass structure, it can capture the coexistence of tetrahedral and octahedral Ge environments within the amorphous network of 225GST, as previously reported from ab initio molecular dynamics simulations[20]. Consequently, the complex local environments coexisting in the glass structure can be successfully reproduced; hence, glass models generated with the GAP potential are not missing any of the atomic environments believed to be present in amorphous 225GST[24].

In this work, an ensemble of 315-atom supercell models of amorphous 225GST was simulated; this cell size corresponds to a compromise between the size of the glass model, the computer time required to achieve representative statistics for the defect states, and the feasibility of performing spin-polarised hybrid-DFT calculations. The quality of the generated amorphous 225GST models, obtained using classical MD with the GAP potential, was validated by comparison of the atomic structure of the simulated glasses with experimental and previous DFT-modelling data. The total radial distribution function (RDF), averaged over the 30 samples, is shown in Supplementary Fig. 1, and it agrees well with the RDF calculated for a 315-atom structural model of amorphous 225GST, generated by ab initio molecular dynamics simulation[26]. In addition, the total X-ray structure factor for the simulated glasses was computed, and is shown in Supplementary Fig. 2. Comparison with an experimental X-ray structure factor[27] shows very good agreement, indicating that the glass models are indeed representative of amorphous 225GST. A quantitative assessment between the experimental and modelling data, presented in Supplementary Note 2, further highlights the very good agreement with respect to the first and second coordination shells of the glass structures.

**Electronic structure and mid-gap defects.** The 30 glass structures obtained using the GAP interatomic potential were further optimised using DFT with a hybrid functional. The inclusion of a portion of the Hartree–Fock exchange to the PBE approximation

is imperative in order to be able to improve significantly the estimation of the band gap, and hence establish a better description of the electronic structure, which is crucial for the investigation of mid-gap electronic states in amorphous 225GST. Hybrid-functional DFT calculations are essential to characterise the localised defect states in the band gap of glassy 225GST, which then can be ascribed to specific structural features within the amorphous network.

The electronic-structure calculations show an average Kohn–Sham band gap of 0.66 eV for the relaxed ground state. Supplementary Table 1 contains the band gap values for all the thirty glass models. The calculated values of the band gap, ranging from 0.55 to 0.79 eV, agree very well with the experimentally reported values for amorphous 225GST[28,29], ranging between 0.6 and 0.8 eV, as well as with previous modelling studies[22,30].

Of the 30 amorphous 225GST models studied here, in 11 of them, corresponding to ≈37% of the glass models, in-gap electronic states emerged. Of the 11 modelled systems with in-gap states: 9 models have a well-defined mid-gap defect electronic state; 5 models have a shallow defect state; 1 model has two defect states; and 2 models have three defect states in their band gap. The mid-gap states are located, on average, 0.34 eV below the bottom of the conduction band and correspond to unoccupied states. The shallow unoccupied defect states are located, on average, 0.17 eV below the conduction-band edge, in good agreement with experimental measurements[23]. The positions of all in-gap electronic states of each glass model are shown in Supplementary Table 1.

The total and partial electronic densities of states of a glass model (sample #4 from Supplementary Table 1) with a mid-gap defect state is shown in Fig. 1, whereas Supplementary Fig. 3 shows the total and partial electronic densities of states of an amorphous 225GST model (sample #3 from Supplementary Table 1) without any gap states, for comparison. For the sake of completion, the total electronic densities of states for all 30 amorphous 225GST models generated in this work are shown in Supplementary Figs. 4–33.

The degree of localisation of each single-particle Kohn–Sham state in the electronic structure of an amorphous model can be characterised quantitatively by calculating the inverse participation ratio (IPR). This method has been previously used to characterise the localisation of vibrational and electronic states in amorphous materials[31–34]. The IPR spectrum near the band edges for the 225GST glass model #4 from Supplementary Table 1, that has a mid-gap electronic state, is shown in Fig. 2. The IPR for delocalised states has values between 0.01 and 0.02, whereas partially localised electronic states correspond to IPR values ranging from around 0.03–0.06. One can clearly observe that the mid-gap defect corresponds to a fairly strongly localised state, with an IPR value of ~ 0.09. In addition, the IPR analysis demonstrates that there are a couple more localised states at the bottom of the conduction band, whereas a few partially localised tail states appear at the top of the valence band of this model of amorphous 225GST.

Hence, the mid-gap states are rather localised in the periodic cell of the amorphous 225GST models. Direct inspection of the wavefunctions can reveal the complex atomic motifs that host these electronic states. Analysis of the molecular orbitals for all the glass models that have a mid-gap state reveals that 5-coordinated Ge atoms correspond to the local environment, which is always involved in the production of the mid-gap states. In a more-extended view, by considering second-nearest-neighbour environments, it can be observed that the defect states consist of small groups of Ge atoms that are not necessarily covalently bonded to each other.

The molecular orbitals associated with mid-gap electronic states in two 225GST glass models are shown in Fig. 3. In the left panel (corresponding to glass sample #4 from Supplementary Table 1), three 5-coordinated Ge atoms, which are not bonded to each other, and form a kind of chain, are associated with the localisation of the mid-gap state. Two 3-coordinated Sb atoms and one 2-coordinated Ge atom also contribute to the whole picture of the structural motif for the defect state. The formation of Ge(Sb)–Te–Ge(Sb)–Te fourfold rings leads to the creation of cubic structural motifs, in which they share a Ge atom, resulting in a (rocksalt) crystalline-like environment for the mid-gap electronic state. In the right panel (corresponding to glass sample #5 from Supplementary Table 1), a group of five Ge atoms are

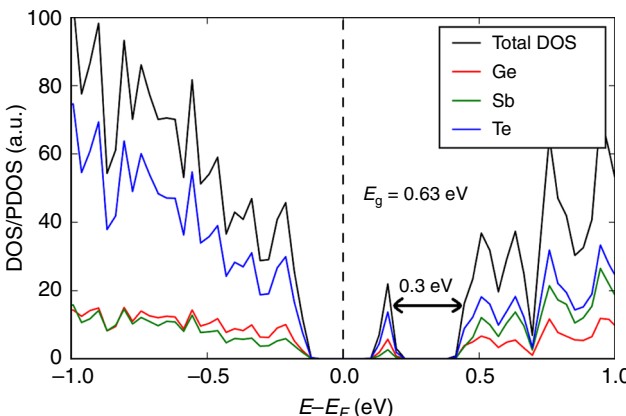

**Fig. 1** Electronic structure of a glass model with a mid-gap defect. Total and partial electronic densities of states (DOS/PDOS) near the top of the valence band and the bottom of the conduction band of a 225GST glass model with a mid-gap defect state in the band gap (glass sample #4 from Supplementary Table 1). A hybrid-functional electronic-structure calculation results in a band gap, $E_g$, of 0.63 eV for the relaxed ground state. The mid-gap electronic state is located at 0.3 eV below the conduction-band minimum. The bottom of the conduction band in the electronic structure is owing to Te and Sb states. In contrast, the mid-gap defect state, which is also an unoccupied state, is dominated by Te-atom states with some contribution from Ge-atom states, whereas a Sb contribution is almost negligible. The Fermi level lies deep in the band gap, between the mid-gap electronic state and the valence-band maximum

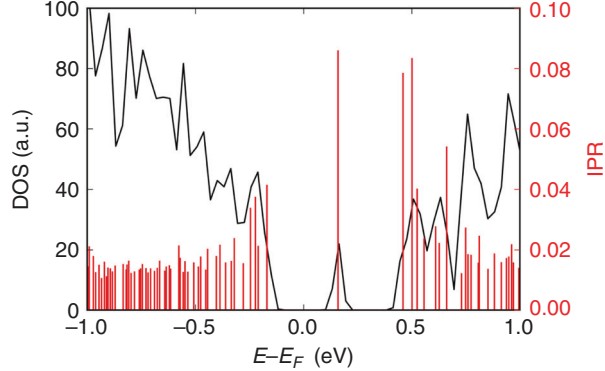

**Fig. 2** Localisation of a mid-gap electronic state. Total electronic density of states (DOS), highlighted with a black solid line, near the top of the valence band and the bottom of the conduction band of an amorphous 225GST model with a mid-gap electronic state (glass sample #4 from Supplementary Table 1). The corresponding inverse participation ratio (IPR) values for the Kohn–Sham orbitals, highlighted with red spikes, reveal strong spatial localisation of the mid-gap defect state

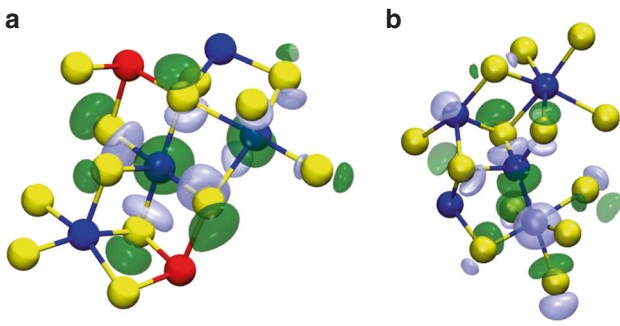

**Fig. 3** Local atomic structure of mid-gap defects. Atomic structures and molecular orbitals of two representative examples of mid-gap electronic states identified in the generated amorphous 225GST models, corresponding to: **a** glass model #4; and **b** glass model #5 from Supplementary Table 1. A 5-coordinated Ge atom is the structural precursor of the local environment, which is associated with the localisation of the mid-gap defect within the amorphous network. Consideration of second-nearest neighbours reveals the formation of small Ge groups, which can be over- and under-coordinated, as well as bonded or non-bonded to each other. In both cases, it can be seen that a crystalline-like environment within the glass structure is the structural motif, which hosts the mid-gap electronic state. Ge atoms are blue, Sb are red and Te are yellow. The light green and blue isosurfaces, in both configurations, depict the molecular-orbital wavefunction amplitude of the mid-gap defect states and are plotted with isovalues of $+0.015$ and $-0.015$, respectively

associated with the localisation of the mid-gap state. Specifically, one 2-coordinated, two 4-coordinated, one 5-coordinated and one 6-coordinated Ge atoms create the structural configuration that hosts the defect state. Two very interesting observations related to this pattern are: (a) a 5-coordinated Ge atom with a Ge–Ge bond next to a cube; and (b) the formation of an odd-numbered, in particular a 5-membered, ring structure within the defect environment.

Zipoli et al.[6] reported that the majority of the defect states in the band gap of models of amorphous GeTe are produced by clusters of Ge atoms close to each other, in which at least one Ge atom is over- or under-coordinated. In some cases, Ge–Ge bonds are present, whereas in some other cases, the Ge atoms are not bonded to each other. In addition, GeTe cubes, not properly aligned and sharing a Ge atom, were also identified as possible structural motifs that can host defect states in the same amorphous material. Zhugayevych and Lubchenko[35] reported specific motifs, which contain over- and under-coordinated atoms, as well as a five-membered closed ring, as structural patterns, which can host mid-gap states in amorphous arsenic selenide. Finally, Gabardi et al.[36] discussed the formation of defects in models of amorphous GeTe which are localised in a kind of Ge–Ge chain-like structure.

**Local atomic environments**. In amorphous 225GST, different local environments, with various atomic coordinations and degrees of ordering, can coexist inside the glass. The local environment of each atomic species in the thirty 315-atom 225GST glass models generated in this work was described by using the smooth overlap of atomic potentials (SOAP) structural descriptor[37] (see Supplementary Note 3 for details). Interatomic distances up to second-nearest neighbours, corresponding approximately to distances 5.5 Å away from the central atom, were considered in this SOAP analysis. The high-dimensional SOAP data can be embedded, with metric multidimensional scaling algorithms[38], into two-dimensional maps in which they can be visualised and interpreted accordingly. The configurational

map shown in Fig. 4a reveals that the SOAP data form three well-defined clusters, corresponding to Ge, Sb and Te local environments. The map is also correlated with the coordination numbers, since environments away from the centre of each cluster correspond to under-/over-coordinated atoms. Hence, it can be observed that, within the Ge cluster, 5-coordinated atoms, which are associated with the localisation of the mid-gap defects, are closer to the edges of the cluster.

The average coordination numbers around Ge, Sb and Te atoms, calculated from the 30 amorphous 225GST models by using a geometrical bond-distance cutoff of 3.2 Å, were found to be 4.2, 3.6 and 2.9, respectively. These results are similar to those reported by ab initio molecular dynamics simulations of amorphous 225GST[22,26,39], as well as being in very good agreement with our recent work[24]. The local coordination environments for all the atomic species in our modelled systems were also calculated, based on information from the charge-density distribution and by using the Electron-Localisation Function (ELF)[40]. The coordination-number histograms obtained from the ELF analysis for the local environments of the three atomics species in the glass models are shown in Supplementary Fig. 34, whereas the average coordination numbers around Ge, Sb and Te atoms, calculated from the thirty glass samples using the ELF analysis, were found to be 3.7, 3.3 and 2.7, respectively (see Supplementary Note 4 for a comparison between the two approaches).

The distribution of local environments around the Ge atoms in the glass models with 3, 4, 5 and 6 other atoms in nearest-neighbour positions, is shown in Fig. 4b. More than 50% of the Ge atoms were found to have four other atoms as first-nearest neighbours within the amorphous network. However, one can see that the proportion of under- and over-coordinated Ge atoms in the glass models is quite significant. About 33% of the Ge atoms are 3-coordinated, whereas 18% of the Ge atoms have five or six atoms in their nearest surroundings, highlighting the broad range of Ge local environments in the glass structure. It should be noted that bond formation between two nearby atoms is considered as long as the interatomic distance between these two atoms is less than or equal to a cutoff distance of 3.2 Å. In Fig. 4c, the proportions of five- and six-coordinated Ge atoms bonded to only Te atoms or to Te atoms and at least one other Ge atom, were calculated, and they reveal that atomic configurations of 5-coordinated Ge atoms bonded to only Te atoms, or with one Ge–Ge bond, are defect configurations, which can be found in the simulated glasses in considerable amounts. Moreover, the ELF analysis demonstrated that 5-coordinated Ge local configurations are evidently present as a structural motif associated with localisation of mid-gap defects in the amorphous network of the simulated models.

The simulated Ge local environments agree well with the proposed geometry of Ge atoms in amorphous 225GST from X-ray absorption fine-structure and X-ray absorption near-edge spectroscopy measurements[41], as well as from high-energy X-ray and neutron-diffraction experimental studies[19]. Similar results on the distribution of the Ge local geometries in the 225GST glass were also reported from ab initio molecular dynamics simulation studies of amorphous 225GST[20–22,39,42]. The effects of different DFT-schemes, reported in the literature[43–46], on the structure of amorphous 225GST are discussed in Supplementary Note 5.

The SOAP analysis and the respective configurational map, presented in Supplementary Fig. 35 and discussed in Supplementary Note 6, indicate that the atomic structures of the different amorphous models generated in this work are quite similar, with respect to the first- and second-nearest neighbours. However, in those glass models with defect states, the amount of 5- and 6-coordinated Ge atoms is larger than in the glass models

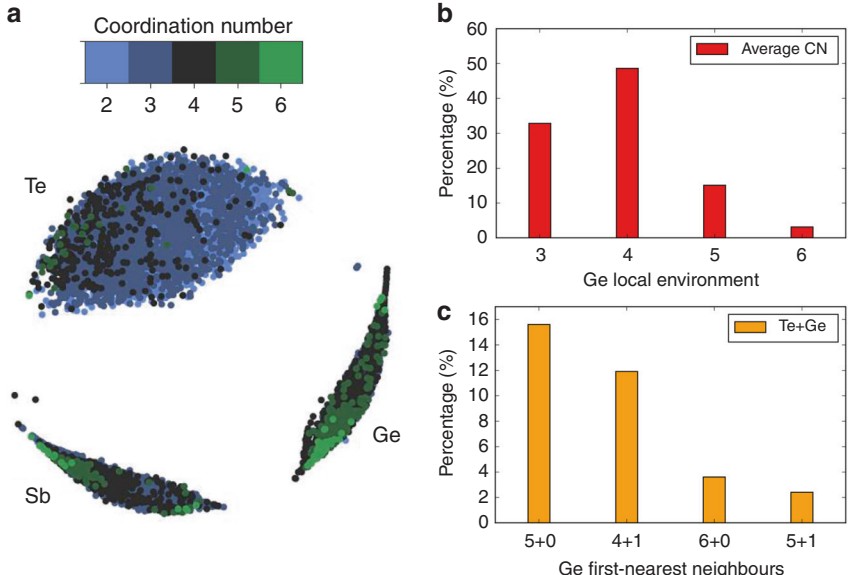

**Fig. 4** SOAP analysis of the coordination environments in the glass models. **a** Two-dimensional SOAP map of the Ge, Sb and Te local environments for each atomic species in the amorphous 225GST structures, generated in this work, with two (light blue), three (dark blue), four (black), five (dark green) and six (light green) other atoms in the nearest-neighbour positions. Environments away from the centre of each cluster correspond to under-/over-coordinated atoms inside the glass network. **b** Local environments of Ge atoms with 3, 4, 5 and 6 other atoms in the nearest-neighbour positions in the glass structure averaged over the thirty 225GST glass models. Despite the four-coordinated local environment being the most-favourable geometry that can be found in the glass, under- and over-coordinated Ge atoms are also possible geometrical arrangements within the amorphous network. **c** About 16% of the 5-coordinated Ge atoms were found to be 5-coordinated to only Te atoms inside the glass structure, whereas ~12% were coordinated to four Te atoms and one other Ge atom, corresponding to two structural motifs related to the localisation of the mid-gap electronic states. The distribution of coordination numbers around the atoms was calculated by using a geometrical bond-distance cutoff of 3.2 Å

without defect states, as shown in Supplementary Fig. 36, highlighting a structural difference between the two types of models, which, even though it is relatively subtle, is significant for the localisation of the defects.

It can be noted that the glass models show also some subtle differences from an energetic point of view. From Supplementary Table 2 and Supplementary Fig. 37, it can be observed that the glass models with additional electronic states in the band gap have a higher total energy, on average, than the 225GST models with a clean band gap. The time-dependent resistance drift in phase-change memory devices has been ascribed previously to the spontaneous structural relaxation (ageing) of the amorphous state created via the melt-and-quench process[47,48]. Experimental studies on glassy GeTe indicated a shift of the in-gap defects and band edges away from the Fermi level, simply owing to a stretching of the density of states, which also leads to a movement of the Fermi level nearer to mid-gap[14,15,49,50]. First-principles calculations on amorphous GeTe indicated also a correlation between the resistance increase and the annihilation of structural defects responsible for localised electronic states, which results in an increase of the band gap of the glass and occurs via a series of collective rearrangements of the defect complexes[6,48]. Based on this perspective, the effect of ageing results in the amorphous phase evolving with time towards an energetically favourable, more "ideal" glass state[47,48]. Our calculations could be taken to support this view and to provide some indication of the effect in amorphous 225GST, as those models among the 30 simulated 225GST glass models with in-gap electronic states were found to be energetically less favourable compared to the models that do not have defect states in their band gaps. Hence, upon structural relaxation, one would expect the in-gap states to disappear gradually with time, in order to reduce the overall free energy of the system, thus resulting in more energetically stable structures and hence leading to an increase of the resistance, as the glass

becomes more similar to some "ideal" amorphous configuration. However, we note that the exact nature of such an "ideal" amorphous state of 225GST requires further investigation.

Five-coordinated Ge atoms correspond to a defective local environment within the amorphous network, which by default means that it is not a dominant configuration. The two-dimensional SOAP map presented in Supplementary Fig. 38 and discussed in Supplementary Note 7 indicates that 5-coordinated Ge atoms are able to participate in crystalline-like atomic structural motifs, which, in turn, can host the mid-gap electronic states of the amorphous material. However, this does not imply that all these 5-coordinated Ge atoms will result, a priori, in a crystalline-like structural pattern, which was found to be associated with the localisation of the defect, as this structural motif appears to be more complex than that, as shown in Fig. 3. Based on the percentage (≈35%) of the glass models that have mid-gap electronic states in their band gap, and in conjunction with the size of the modelled system (315 atoms), we would expect to find, on average, one such crystalline-like environment for every ~900 atoms.

**Modelling a 900-atom glass**. This estimate for the defect probability obtained from the statistics for the database of the 30 315-atom glass models was verified after corresponding hybrid-DFT calculations for a 900-atom amorphous 225GST model generated using the GAP potential. The total electronic density of states of this 900-atom glass structure, together with the calculated IPR spectrum near the band edges, shown in Fig. 5a, reveals the existence of several, deep (mid-gap) and shallow, localised unoccupied electronic states in the band gap of the glass. The hybrid-DFT electronic-structure calculations show a band gap of 0.63 eV for the relaxed ground state of this larger amorphous model, whereas the mid-gap electronic state is located at 0.37 eV

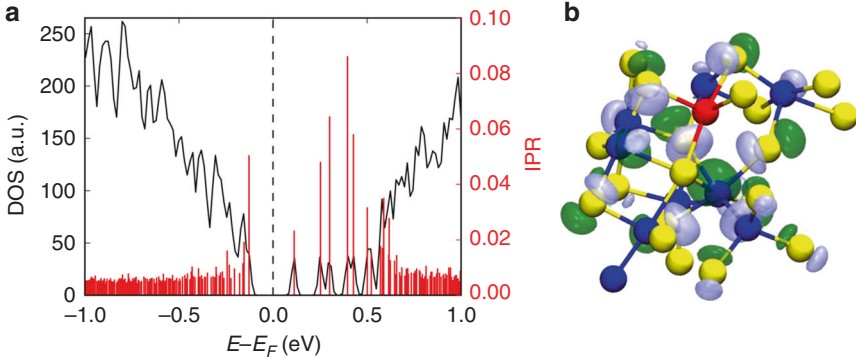

**Fig. 5** Electronic-structure calculations for a 900-atom amorphous 225GST model. **a** Total electronic density of states (DOS), black solid line, near the top of the valence band and the bottom of the conduction band for an amorphous 225GST model of 900 atoms. A hybrid-functional electronic-structure calculation produces a Kohn–Sham band gap of 0.63 eV for the relaxed ground state. Several localised, unoccupied electronic states can be identified in the band gap of the glassy model. The mid-gap electronic state is located at 0.37 eV below the conduction-band minimum. The corresponding inverse participation ratio (IPR) values for the Kohn–Sham orbitals, red spikes, show the strong spatial localisation of the mid-gap and shallow defect states. The Fermi level lies deep in the band gap of the glass model, at a position between the lowest mid-gap electronic state and the valence-band maximum. **b** Atomic structure and molecular orbital associated with the mid-gap electronic state identified in the band gap of the 900-atom glass model. Five- and six-coordinated Ge atoms are involved in the localisation of the mid-gap defect state, whereas the presence of four-fold connected rings leads also to the creation of a cubic structural pattern in the configuration of the defect. Ge atoms are blue, Sb are red and Te are yellow. The light green and blue isosurfaces depict the molecular-orbital wavefunction amplitude of the mid-gap defect state, and are plotted with isovalues of +0.015 and −0.015, respectively

below the bottom of the conduction band. Visualisation of the wavefunction for the mid-gap defect state present in this modelled system, shown in Fig. 5b, reveals an associated atomic geometry with a close resemblance to those that host the mid-gap electronic states in the 315-atom series of glass models. It can be observed that a crystalline-like atomic environment is again hosting the mid-gap defect state, whereas the involvement of 5-coordinated Ge atoms can be identified in the associated defect-state localisation.

The hybrid-DFT electronic-structure calculation of the 900-atom glass model supports the statistical accuracy of the database of the 30 315-atom 225GST amorphous models, since, from them, mid-gap defect states and their corresponding atomic environments were also traced inside the glass structure. Moreover, there is no previous study in the literature, to the best of our knowledge, that has ever reported geometry optimisation for such a large system size of an amorphous structure using non-local functionals.

In the 900-atom glass model, the defect states other than the deepest unoccupied electronic state in the band gap are also localised among groups of 5-coordinated Ge atoms, corresponding to structural configurations, which are quite similar to that observed for the mid-gap defect environment. Nevertheless, these defects appear not to have a pronounced crystalline-like extended structural motif. In the 315-atom glassy models, the shallow defects, are located, on average, at ≈0.17 eV below the conduction-band edge, in good agreement with experimental observations (≈0.2 eV)[23], and are localised on 3- and 4-coordinated Ge atoms, whereas more Sb atoms participate in the localisation of the electronic states. It can be noted that the atomic configuration associated with these shallow defects was observed to have a closer resemblance with the atomic environment which characterises the bottom of the conduction band in those glass models that do not have mid-gap defect states.

**Local pressure and defects**. The underlying physical mechanism for the origin of the resistance drift in the amorphous state of phase-change memory materials has been correlated previously with the stress induced by the phase-change itself[51,52]. Nevertheless, this is a debatable view, as Rizzi et al.[53] reported that the

resistance drift in phase-change materials is not impacted by stress. In this work, the local virial stress was calculated for every amorphous 315-atom 225GST model, in order to gain an insight into the mechanical stress in the glass structures at the atomic scale. The stress tensor for each atomic species in a glass model was calculated by using the GAP framework, exploiting the fact that, in a local potential like GAP, the total energy can be written uniquely as a sum of local terms, and hence it is more interpretable for each atom in the simulation cell[54]. Subsequently, the trace of the stress tensor was computed to obtain a value for the local pressure for each atom in the glass, and the results are shown in Fig. 6 for the 225GST glass model #5 from Supplementary Table 1, that has a mid-gap defect state. A more-detailed description of the calculation of the local stress in the amorphous structures is presented in Supplementary Note 8. The total residual stress ranges between 1.3 and 1.7 GPa in the glass models after the geometry optimisations with hybrid-DFT calculations. A comment about the fixed-volume approach followed in this work to generate glasses, and the resulting residual stresses in the cell, is given in Supplementary Note 9.

Figure 6 also shows, enlarged, the local stress in the atomic environment in which a mid-gap state is localised within the amorphous network. It can be observed that the distribution of the stress in the entire computational cell is rather inhomogeneous, even though the average stress experienced by the amorphous network is low. The local stress in the structural motif in which the mid-gap state is localised was found to be particularly low, with a value even smaller than the average stress in the glass model, which can be expected for a crystalline-like atomic environment.

**Electron trapping calculations**. Mid-gap defects in amorphous semiconductors are believed to serve as either effective recombination centres or carrier traps, depending on whether the defect possesses a small or a large difference in capture rates of electrons and holes, respectively[55]. In addition, localised unoccupied and occupied electronic states in the band gap of an amorphous semiconductor can give rise to possible charge trapping[33]. The IPR spectrum in Fig. 2 suggests that there can be several electron-trapping sites within the amorphous network of the

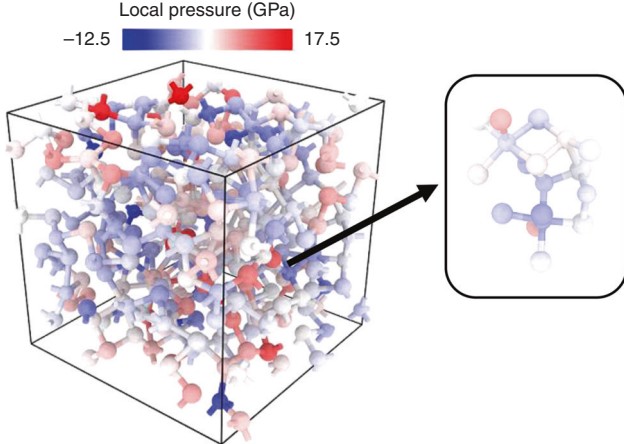

Local pressure (GPa)

−12.5 ▬▬▬▬ 17.5

**Fig. 6** Distribution of the virial stress in a glass model with a mid-gap defect. Left panel: local pressure calculated with the GAP framework for each atomic species in the glass structure for an amorphous 225GST model with a mid-gap electronic state (glass sample #5 from Supplementary Table 1). A colour palette between blue and red indicates the value of the local pressure for each atom in the glass model, which is ranging from −12.5 to 17.5 GPa. The distribution of the local stress is inhomogeneous in the periodic cell; nevertheless, the average local pressure is low in the glass model. Right panel: computed local pressure in the atomic configuration in which the defect state is localised within the glass network. The atomic environment which houses the mid-gap defect state also experiences a low stress, which is in accordance with its crystalline-like structure

225GST models, as there are several unoccupied states with large IPR values. The mid-gap defects identified in our simulated 225GST glasses give rise to highly localised, unoccupied electronic states in the band gap. In order to investigate if the mid-gap defects can capture an electron and, furthermore, to find the degree of electron localisation, requires these states to be occupied, followed by a full geometry optimisation.

Therefore, an extra electron was added to the relaxed ground state of those glass models, having a mid-gap electronic state in their band gap, and the geometry of each system was then re-optimised. The electronic-structure calculations revealed that all the mid-gap defects present in the relevant 225GST glasses are able to capture the extra electron. The electronic density of states of one of the simulated glasses (model #14 from Supplementary Table 1), in which an electron is trapped, is shown in Fig. 7a. The mid-gap electronic state becomes an occupied molecular orbital, and the average position of this state, from all the glass models containing a mid-gap defect state, is ≈0.54 eV below the bottom of the conduction band, bringing this state very close to the top of the valence band, and hence indicating a relatively deep electron trap. The positions of the electron traps in the glass models, having a mid-gap defect state, are shown in Supplementary Table 1.

An analysis of the spin-density distribution, for seven glass models with mid-gap electronic states, reveals a strong localisation of the extra electron in a structural motif, analogous to that responsible for the mid-gap defects in the amorphous 225GST models. An example of electron-centre formation in one of the amorphous structures (glass model #14 from Supplementary Table 1) is shown in Fig. 7b. The extra electron is localised amongst a group of four Ge atoms within the glass structure, with two of them being 5-coordinated to Te atoms. The formation of Ge–Te–Ge–Te fourfold connected rings highlights the crystalline-like atomic environment of the electron trap.

Geometry optimisation shows that electron trapping causes some distortion in the local atomic structure associated with the mid-gap defect states. The electron localisation leads to weakening of some of the Ge–Te bonds in the configuration of the defect, which become longer by 0.1–0.2 Å. On the other hand, the Ge atoms move closer to each other by ~0.1 Å. The average displacement of the Te atoms in the defect configuration, between the relaxed ground state and the relaxed electron-trap state, was calculated to be ≈0.14 Å, whereas the average Ge-atom displacement was found to be ≈0.13 Å. It can be noted that such structural relaxation is characteristic of electron-centre-type states in amorphous semiconductors.

A different view about the resistance drift, which does not involve structural-ageing effects, can be drawn, inspired by the unique way that amorphous 225GST is made in PCRAM devices. Instead of simply quenching a liquid formed by the external thermal melting of a crystal, the thermal energy for melting in a PCRAM cell is provided internally by the Joule heating associated with the application of a RESET voltage pulse. The I–V characteristics of 225GST are non-linear and the non-Ohmic increase in current is owing to electric-field-assisted carrier-generation processes[11]. Hence, when the amorphous phase of 225GST forms in a PCRAM cell, it is in the presence of a very high electron-hole current density. Thus, under such conditions, most of the localised gap-state trapping centres would be expected to become filled during the RESET process itself[9]. Thereby, the electronic condition of the material immediately post-RESET is equivalent to that an amorphous semiconductor that can be found after steady-state pulsed optical excitation and before the following transient photocurrent decay. Electrons/holes are subsequently thermally de-trapped; an applied read voltage therefore can produce a time-dependent current, which can be interpreted as a time-dependent resistance. Consequently, this type of behaviour can be used to explain the very short-time increase in resistance evident in amorphous 225GST immediately after the reset pulse. It is noted that this approach has been previously used to describe the long-time photocurrent decay exhibited by amorphous silicon[56].

This alternative electronic mechanism for the origin of the resistance drift is based on the fact that in-gap localised states in the amorphous phase can serve as charge-carrier traps, and the long-time resistance-drift effect is related to the slow deep-trap release and subsequent recombination with charge carriers. From the electron-trapping calculations in this study, we have highlighted that the unoccupied mid-gap defect electronic states in the band gap of the amorphous 225GST models are able to capture electrons, leading to deep-trap states with strong electron localisation in the structural motif of the defect. This observation works in favour of the view that the resistance drift in amorphous 225GST might be caused by electron/hole trapping during the electron/hole-injection process accompanying the RESET process and creation of the glass.

In conclusion, understanding the structural origin of resistance drift in phase-change materials is indispensable for its elimination and subsequent development of multi-level, multi-bit storage devices. The atomistic modelling presented in this work links the electronic density of states to the structure of the glass, and establishes the structural motifs which are associated with the localised mid-gap electronic states of the prototypical phase-change memory material, 225GST. The results provide significant insight, as well as a prospect of mitigating resistance drift in phase-change memory devices by suitable material engineering to control the distribution of localised defect states within the band gap of the amorphous phase, thereby facilitating the realisation of multi-level storage operation in future PCRAM devices.

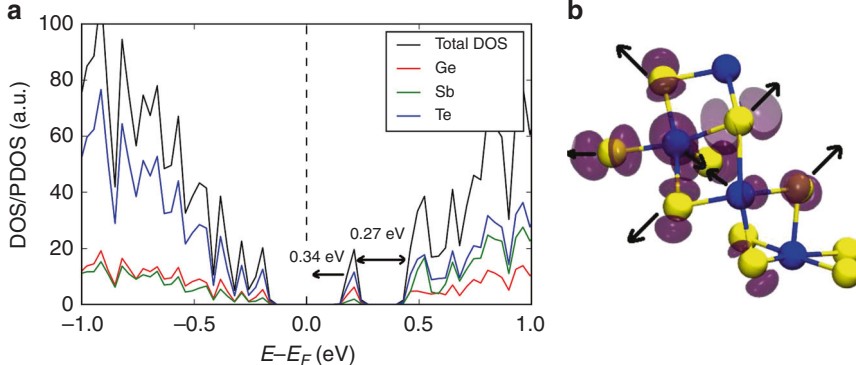

**Fig. 7** The mid-gap defect states as deep electron traps. **a** Total and partial electronic densities of states (DOS/PDOS) of an amorphous 225GST model (glass sample #14 from Supplementary Table 1) with a mid-gap defect state located at 0.27 eV below the bottom of the conduction band. An extra electron can occupy the mid-gap electronic state, and this state shifts by 0.34 eV towards the valence-band maximum following geometry relaxation, as indicated by the arrow. The mid-gap localised defects in the amorphous 225GST models are able to capture electrons and correspond to deep traps in the band gap of the material. **b** Spin-density distribution and atomic geometry of an electron trap in the glass model #14 from Supplementary Table 1, which has a mid-gap electronic state. The extra electron is well localised on a small cluster of Ge atoms within the glass structure. Five-coordinated Ge atoms are associated with the electron localisation and the structural motif of the electron defect centre has a crystalline-like character, which characterises the mid-gap defect states identified in the amorphous models. Arrows indicate the direction of atomic displacements following electron trapping. Ge atoms are blue and Te atoms are yellow. The isovalue of the spin density (light-purple isosurface) is equal to 0.001

## Methods

**Molecular dynamics simulations**. Thirty model structures of amorphous 225GST were generated using classical MD with periodic boundary conditions. The total number of atoms in each periodic cell was 315, with the individual number of atoms being Ge = 70, Sb = 70 and Te = 175. Initially, the atoms were placed randomly in a cubic simulation box, with the cell size calculated from the experimental density (5.88 g/cm³)[57]. A machine-learned interatomic potential for amorphous 225GST[24] was employed for the classical MD simulations. The force field was obtained within the GAP framework[58] by fitting a database of configurations, which were evaluated with DFT.

The MD simulations were performed using the LAMMPS code[59] and the glass structures were generated following a melt-and-quench approach. The canonical ensemble (constant number of particles, volume and temperature, or NVT) was applied to keep the density of the simulated amorphous model close to the experimental value. The Nosé–Hoover thermostat, with a relaxation constant of 40 fs, was chosen to control the temperature fluctuations. A time-step of 1.0 fs was used to integrate the equations of motion during the MD simulations. For each glass model, the initial configuration was heated up at 3000 K with a 30 ps MD run to ensure that the system was melted. The molten structure was subsequently cooled down to 1200 K and equilibrated using first the NVT and then the NVE (microcanonical) ensembles with 30 and 10 ps MD runs, respectively, in order to obtain a well-equilibrated liquid structure at this temperature. The system was then linearly cooled down to 300 K at a quenching rate of −15 K/ps. At 300 K, the glass structure was equilibrated for 30 ps, followed by a 10 ps MD run with the NVE ensemble, to collect structural data. Finally, the temperature of the system was brought down to around 0 K at a quenching rate of −15 K/ps. The computational scheme used in this work corresponds to a total simulation time of 190 ps for each model.

An amorphous 225GST model of 900 atoms was also generated by GAP-MD simulation, following the same melt-and-quench protocol used for the 315-atom glass structures. In Supplementary Note 10 is discussed how the molecular dynamics simulations performed in this work can ensure the independence of the generated amorphous models.

**Electronic-structure calculations**. The output glass structures at 0 K from the melt-and-quench simulations were used as input configurations, for the 30 315-atom and the 900-atom amorphous 225GST models, to further optimise their geometry, using DFT implemented in the CP2K code, and to calculate the electronic structures[60]. The CP2K code uses a Gaussian basis set with an auxiliary plane-wave basis set[61]. All atomic species were represented using a double-ζ valence-polarised Gaussian basis set[62] in conjunction with the Goedecker-Teter-Hutter pseudopotential[63]. The plane-wave energy cutoff was set to 400 Ry. The atomic geometry of each glass model was optimised, and the electronic structure was calculated by using the non-local PBE0 functional, with a cutoff radius of 3 Å for the truncated Coulomb operator[64]. The inclusion of the Hartree–Fock exchange provides an accurate description of the band gap and the localised defect states in our 225GST glass models, as was demonstrated in our previous work[65]. The computational cost of hybrid-functional calculations can be reduced by using the auxiliary density-matrix method[66] (see Supplementary Note 11 for details), as employed in previous modelling studies of amorphous materials[67,68]. The

Broyden–Fletcher–Goldfarb–Shanno algorithm was applied in the geometry optimisations and the forces on atoms were minimised to within 0.023 eV/Å. Electron trapping was modelled by injecting an extra electron in the relaxed ground-state glass structure and minimising the energy with respect to the atomic coordinates. The same non-local functional, as well as the same set-up, were used for the geometry optimisations of the glass systems with the extra electron.

## Data availability

Data supporting the observations in this study are available from the corresponding author upon reasonable request.

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

## Acknowledgements

This work was supported by the UK Engineering and Physical Sciences Research Council (EPSRC) grant EP/N022009. F.C.M. acknowledges the EPSRC Centre for Doctoral Training in Computational Methods for Materials Science for funding under grant number EP/L015552/1. Via our membership of the UK's HEC Materials Chemistry Consortium, which is funded by EPSRC (EP/L000202, EP/R029431), this work used the ARCHER UK National Supercomputing Service (http://www.archer.ac.uk). K.K. acknowledges the use of the UCL Grace High Performance Computing Facility (Grace@UCL), and associated support services, in the completion of this work.

## Author contributions

K.K. designed the research study and the concept was developed by all authors, K.K. carried out the simulations, K.K., F.C.M. and T.H.L. analysed the simulation results, K.K. and F.C.M. constructed the figures, K.K., F.C.M., T.H.L. and S.R.E. interpreted the data, and K.K. and S.R.E. wrote the manuscript.

## Additional information

**Competing interests:** The authors declare no competing interests.

