## [Peer Review File · Nature Communications]

Reviewers' comments:

Reviewer #1 (Remarks to the Author):

In the present work, the authors studied the origin of the gap states in the amorphous phase-change material Ge₂Sb₂Te₅. First of all, twenty amorphous structures are generated using classical molecular dynamic simulations combined with machine-learned potential derived in previous work of the authors, in order to obtain reliable statistics in the following DFT studies. The authors found that the 5-coordinated Ge atoms are related to the gap states in DOS profile. These 5-coordinated Ge atoms and other neighboring atoms form a crystalline-like environment with low local stress. Furthermore, the electron-trapping property of these gap states is studied. Overall, by using very comprehensive calculations, the present work presents insights into the nature of the gap-states in amorphous Ge₂Sb₂Te₅. However, revisions are needed before the publication of the present manuscript, and in particular, the following issues should be addressed.

1. As mentioned in the manuscript, to obtain sufficient statistics, 20 independent amorphous structures are generated. However, the authors did not say how to ensure the 'independency' of the structures. Based on what kind of rules the structures are picked from the MD simulations and how different the atomic structures are, should be mentioned, which are crucial for the statistics.
2. The gap states are traced to a crystalline-like environment containing 5-coordinated Ge. How many of this kind of atomic cluster can be found in one typical sample with gap states?
3. Unoccupied gap-states are found in 7 samples of the 20 structures. The percentage of having the gap states is 35%, which is quite low. The authors mention that gap state is associated with crystalline-like environment having low local stress. Is this environment energetically stable? If so, why it is missing in most of the samples? Can the authors comment on this?
4. More discussions are needed as the authors try to link the gap-states or 5-coordinated Ge atoms to the resistance-drift of the glass structure. It is unclear in the present manuscript.
5. In this work, glass structures obtained from MD simulations are further optimized using DFT with hybrid functional. It will be very meaningful for the reader if the authors can show the difference between the relaxed structures based on the hybrid functional and those based on GGA/LDA which is normally used in the stage of structure optimization.
6. In the Method section, the authors may describe the calculation procedure of the local stress in the amorphous structure with more details.

Reviewer #2 (Remarks to the Author):

This manuscript reports on a computational study of the amorphous state of Ge₂Sb₂Te₅ (GST), an important phase-change material employed in non-volatile phase-change RAMs. Phase-change materials display fast and reversible transformations between two phases (a crystalline and an amorphous phase) that show different transport properties. In standard phase-change RAMs, two-state memory cells are used. Employing multi-level cells based on intermediate resistance levels would enable an increase in storage density. However, the development of multi-level cells is hampered by the resistance drift phenomenon, which is due to the ageing of the amorphous state. The goal of this work is the characterization of the geometric and electronic properties of the midgap states of amorphous GST, which affect the transport properties of this phase. The authors have used a machine-learning Gaussian approximation potential (GAP) (introduced in a previous work, Ref. 29) fitted to DFT simulations to model the interaction between the atoms. Classical MD simulations based on this potential have been used to generate amorphous models. The classical simulations have been complemented with DFT calculations with hybrid functionals to determine the band gaps and characterize the electronic states. The main finding is that localized midgap states exist in crystalline-like local atomic environments containing 5-fold coordinated Ge atoms.

The paper is well written and contains some interesting analyses of the structure/electronic

properties (SOAP structural descriptors, effects of an extra-electron on the geometry of the defects hosting the midgap states). However, I do not recommend publication of this manuscript in Nature Communications, for a few reasons. The first reason is that the manuscript does not advance substantially the field, since it does not shed light on the resistance drift phenomenon. The authors do not propose convincing models to describe the drift. Are the crystalline-like atomic environments expected to disappear during the relaxation of the amorphous network? If yes, why? Should the disappearance of these environments lead to a decrease in energy?

The second reason is that I have concerns about the validity of the findings reported in this work. My main concern is related to the accuracy of the GAP: in their previous paper (Ref. 29), the authors have shown that the GAP amorphous models have a smaller number of tetrahedral Ge (5%-11%) than the DFT models (20%-30%) (note that, in the present work, it is instead claimed that "more than 50% of the Ge atoms are tetrahedrally coordinated within the amorphous network". I assume that, by "tetrahedrally coordinated", it is meant here that the Ge atoms have 4 nearest neighbors; if not, there would be an alarming discrepancy between this work and Ref. 29). In Refs. 27 and 46, it was shown by DFT simulations and classical simulations with neural-network potentials, respectively, that tetrahedral Ge and wrong GeGe bonds (which stabilize tetrahedral Ge) play a crucial role in the ageing process of GeTe - the parent phase of GST. In these works, relaxation of amorphous GeTe was induced by chemical substitution and metadynamics, respectively: the relaxation process was shown to be driven by the disappearance of Ge-Ge chains and tetrahedral structures. Moreover, in Ref. 46, it was argued that similar chains of wrong bonds (Ge-Ge, Ge-Sb, Sb-Sb) are present in GST and disappear during ageing. Since the number of tetrahedra (and wrong bonds, I guess) in the GAP models deviates significantly from the DFT ones, I doubt that it is possible to extract useful information on the ageing process from the GAP models.

I also wonder why the authors have investigated such small system size (~ 300 atoms). Since the authors employ classical GAPs, they could have afforded to study significantly larger models (an order of magnitude larger, I guess). I understand that they also performed DFT simulations with hybrid functionals afterwards, however I don't think this is a good reason not to generate larger models. Even if hybrid DFT calculations are not feasible, it is still very interesting to analyze the structural properties of large GAP models and determine the occurrence of different structural motifs to improve the statistics. Did the authors compare the hybrid-functional calculations with standard GGA simulations? The latter functionals can yield spurious charge delocalization (and definitely underestimate the gap), but, maybe, they are good enough to grasp the localization of the states hosted by the crystalline-like atomic environments. If this is the case, one could combine the generation of large-scale GAP models with GGA simulations of the electronic properties. Furthermore, I do not understand why the quenching rates are so high (10^{13} K/s). The use of machine-learning potentials should also enable much longer simulations. Studying large models on long time scales should be the main reason for the development of a classical potential! Assessing the effects of quenching rates on the structure of amorphous GST is important. What happens if quenching rates of order 10^{12} K/s are employed? Would the systems crystallize during quenching? If yes, is this behaviour due to finite-size effects or to the GAP potential? Is it possible that the crystalline-like atomic environments are crystalline precursors, which reflect the extreme tendency of the system to crystallize?

Reviewer #3 (Remarks to the Author):

Phase change materials (PCMs) serve as one of the most promising candidates for next-generation non-volatile phase change memory (PCRAMs) applications with possible applications to revolutionary concepts such as neuromorphic computing. $\text{Ge}_2\text{Sb}_2\text{Te}_5$ is arguably the most archetypal compound amongst this class of systems.

In this work, Kostantinou and co-workers reported a thorough study by means of first-principles (DFT level) and classical (via a developed machine-learned potential) simulations about the

correlation between the time-dependent resistance drift of amorphous GST and its structure. A central focus is dedicated to the origin of the resistance drift identified in mid-gap electronic states in the band gap originated by specific local bonding defective environments.

Resistance drift as well as threshold switching of amorphous PCMs are indeed currently puzzling phenomena to the field. By resolving these puzzles they would deliver tremendous impact to the PCMs community and PCRAMs technology.

In the following I identify few major concerns and points that I suggest to be addressed before a final decision can be reached for this manuscript:

-The machine-learned potential employed for generating multiple GST glass structures is claimed to produce near-DFT accuracy GST models with "excellent" agreement with experimental X-ray structure factors. However, in Fig. S2 only a qualitative sim. vs exp. comparison it is shown with only a direct comparison for the $g(r)$ DFT vs GAP (Fig. S1).

A quantitative assessment of the structure factor comparison sim. vs exp. it would be definitely helpful to sustain the above claims. For instance, a straightforward way could be to use the goodness-of-fit parameter of Wright et. al. [J. Non-Cryst.Solids 71, 295 (1985)], which is widely employed in the glass community exactly for this purpose.

-Moreover, more arguments should also be detailed on the specific choice of the DFT procedure used. In fact, multiple GST glass models have been developed in the last decade by first-principles molecular dynamics (FPMD) with different DFT schemes in terms of xc functional, pseudopotential, etc [Sci. Rep. 6, 25981 (2016); Phys. Rev. Lett. 103, 195502 (2009)] producing in certain cases even better quantitative agreement with exp. data [Phys. Rev. B 96, 224204 (2017); J. Appl. Phys. 113, 134302 (2013)].

-If I correctly understand the local (defective) coordination scenario of the GST species is analyzed using a geometrical bond-distance cut-off: this approach is arguably appropriate for such a complex type of chalcogenide glass. In fact, for GST and other chalcogenides the local coordination analysis has to be combined whether with Electron-Localization Function ELF (Adv. Mater., 29, 1700814 (2017)) or the Wannier centers formalism analyses (Phys. Rev. B 93, 115201 (2016); Phys. Rev. B 96, 224204 (2017)) to confirm that there is actual bonding interaction between the species. It would be instrumental for the reader to see if actually the mentioned defective 5-coordinate Ge atoms show any sort of interaction with all the nominally coordinated surrounding neighbors. And if yes, what the type of interaction they show.

-Regarding the stress tensor calculations, it would be instrumental to know the exact value of the final glass models residual stress. Generally, a value of cell residual stress >1 GPa is an index of the fact that the glass model has not been relaxed sufficiently. Or, in another words, that the simulation is not predicting the density correctly.

-The strategy employed by the authors to investigate the origin of the resistance drift basically follows the scheme employed by Zipoli et al. [Phys. Rev. B 93, 115201 (2016)] with a detailed analysis of the mid-gap electronic states of multiple GST glass models. Although the results shown in this work and their interpretation sound to be comprehensive at this level, to rule out any doubt more statistics is needed. At this stage only #7 configurations have been used to draw the conclusions on the correlation between the mid-gap electronic states and specific local defective coordination environments of GST species. After all, one of the supposedly advantage of having developed for the first time a machine-learned potential for such complex chalcogenide glass system is the possibility to simulate larger systems or produce "good" GST glass models at less expensive computational cost than with purely FPMD/DFT schemes.

Reply to Reviewer #1

Reviewer #1 Q1

As mentioned in the manuscript, to obtain sufficient statistics, 20 independent amorphous structures are generated. However, the authors did not say how to ensure the 'independency' of the structures. Based on what kind of rules the structures are picked from the MD simulations and how different the atomic structures are, should be mentioned, which are crucial for the statistics.

Authors: We would like to clarify that for the simulations presented in our manuscript, each glass model was generated from scratch. We did not quench from the same liquid structure, for example (which is a virtually independent approach), to obtain amorphous structures, nor did we just select snapshots from a trajectory at a specific temperature (300K) to accumulate glassy atomic environments. In our study, for each glass sample, the atoms, initially, were placed in a pseudo-random arrangement (with a different seed in each case) inside a cubic simulation box. Every system was melted at a high temperature (3000K) for sufficiently long time (30ps) to ensure that any memory effect of the initial pseudo-random guess of the atomic positions is lost before the system was equilibrated to the liquid temperature and then subsequently quenched. For the melting (3000K) and liquid (1200K) temperatures, we also used different seeds for the velocity distributions before starting the MD runs for each model. Finally, it is important to realize that each configuration was subjected to independent NVT runs during the MD simulations of the glass samples.

In Fig. 4(a) of the manuscript, we employed SOAP structural analysis in order to describe the Ge, Sb and Te local environments inside our glass structures with respect to the coordination numbers of all the atoms in every glass sample. For the construction of this configurational map, we used the atomic SOAP descriptors to define the coordination local environment of each atom in a glass model. In order to further explore the similarity of the generated amorphous models, we decided to construct a map by using the global SOAP descriptor [1] for the structure of each glass sample as a whole. For comparison purposes, twenty liquid models and an ideal crystalline configuration were added as landmarks, and the results are shown in the configurational map presented in Fig. 1. In this map, each point corresponds to a different structural 225GST model, while the relative distances between the individual points are correlated with the structural similarity of the different samples. It can be observed that regions corresponding to the liquid, amorphous and crystalline structures are clearly separated. However, glass models with in-gap states (highlighted with green points) cannot be easily distinguished from their amorphous counterparts without any defect states present in their band gap (highlighted with red points). Hence, the SOAP analysis indicates

that the atomic structure of the different amorphous models, with respect to the first- and second-nearest neighbours, is quite similar.

Fig. 1. Two-dimensional configurational SOAP map of liquid (blue points) and amorphous structures, with mid-gap electronic states (green points) and without defect states (red points) in their band gap, for the 225GST models generated in this study. The SOAP descriptor for an ideal 225GST crystalline structure (orange point) has been added to the map for the sake of comparison. Distances between two points on the map indicate the degree of structural similarity between the various structures; the closer, the more similar.

Nevertheless, based on the observations from the localization of the mid-gap states within the amorphous network, we decided to look into some particular structural features of the glass samples in order to try to find some differences between the glass models with and without in-gap electronic states, which will help us to identify some sort of dissimilarity. For example, we observed that 5-coordinated Ge atoms correspond to a defective local environment responsible for the localisation of the mid-gap states inside the glass. Hence, we compared the calculated Ge local environments in the glass models with defect states and in the samples with a “clean” band gap. The Ge coordination numbers were computed by using a geometrical bond-distance cut-off and the results are shown in Fig. 2. One can observe that in the glass models with mid-gap electronic states, the amount of 5- and 6-coordinated Ge atoms is larger than that in the glasses without defect states, highlighting a structural difference between the two “types” of samples, which is significant for the localization of the defects. In addition, the number of 3-coordinated Ge atoms is also larger in the glasses with mid-gap states, whereas the amount of 4-coordinated Ge atoms is larger in the models without in-gap states, indicating, overall, the presence of more defective Ge local environments within the amorphous network of the samples with in-gap electronic states in their band gap.

Fig. 2. Comparison of the local environments of Ge atoms with 3, 4, 5 and 6 other atoms in the nearest-neighbour positions between the glass structures without in-gap electronic states (“clean” band gap) and the glass structures with mid-gap defect states in the band gap. Glass models with mid-gap states have more defective Ge environments (3-, 5- and 6-coordinated Ge atoms) within their amorphous network. It should be noted that the coordination numbers around the Ge atoms were calculated by using a geometrical bond-distance cut-off of 3.2 Å.

Moreover, the observation presented in the manuscript, that the mid-gap defects are hosted in a crystalline-like environment inside the glass models, led us to search for potential differences between the modelled systems regarding the size of the voids. The crystalline-like nature of the structural motif related to the localisation of the mid-gap electronic states suggests that larger voids may be formed in the vicinity of the defective environment. Therefore, we calculated the voids distribution in all the glass models generated in this work and the results are shown in Fig. 3. We can observe that in models with a mid-gap state, the average void-size is slightly higher compared to models with a “clean” band gap, which is in accordance with our expectation that larger voids will be present in the glasses with in-gap states.

Fig. 3. Void histograms calculated for all the 225GST amorphous models generated in this study. A log-normal distribution fitting is represented by the solid lines for the distribution of voids in all the glass models, in glasses without in-gap states (“clean” band gap), and in systems which have mid-gap defects, while the dashed vertical lines correspond to the mean void radius size for each case. Models with a mid-gap electronic state in their band gap show a slightly higher average void size compared to the glass samples with a “clean” band gap.

The two observations from the analysis for the Ge local environments and the size of the voids could provide some structural signatures in order to distinguish the glass models with and without in-gap states. However, we truly believe that these differences are relatively subtle, and, even though they can give some sense of structural dissimilarity with respect to the atomic geometry between the two types of glass samples, we would not like to use them in order to claim something definitive regarding our view of the glasses, which could be misleading for the readers.

Revisions: We commented in the Supplementary Information about the independent character of the glass models generated in this work. We added Fig.1 and Fig.2 from above in the Supplementary Information together with some relevant discussion. We commented in the main paper about the similarity of the generated amorphous models and the observed structural differences with respect to the Ge local environments.

Page 13 - paragraph 1.

The modified text reads: “The similarity of the generated amorphous models was explored by using the global SOAP descriptor⁴⁸ for each glass sample. The SOAP analysis and the respective configurational map, presented in **Fig. S35** in the Supplementary Information, indicate that the atomic structures of the different amorphous models generated in this work are quite similar, with respect to the first- and second-nearest neighbours. Nevertheless, it was observed that 5-coordinated Ge atoms correspond to a defective local environment responsible for the localisation of the mid-gap states inside the glass. For this reason, the calculated Ge local environments in the glass models with in-gap defect states were compared to those in the glass samples with a “clean” band gap, and the results are shown in **Fig. S36**. This analysis shows the presence of more defective Ge local environments within the amorphous network of the samples with in-gap electronic states in their band gap. In particular, in glass models with defect states, the amount of 5- and 6-coordinated Ge atoms is larger than in the glasses without defect states, highlighting a structural difference between the two “types” of samples, which is significant for the localization of the defects. However, it can be noted that, even though there is some structural dissimilarity with respect to the atomic geometry between the two “types” of glass samples, the observed structural differences are relatively subtle.”

Reviewer #1 Q2

The gap states are traced to a crystalline-like environment containing 5-coordinated Ge. How many of this kind of atomic cluster can be found in one typical sample with gap states?

Authors: As we already discussed in our response to the previous question, the amorphous models generated in this study are quite similar, while some subtle differences were identified between the glass structures that have mid-gap states and the glass samples without defect states. Five-coordinated Ge atoms correspond to a defective local environment within the amorphous network, which by default means that it is not a dominant environment. From Fig. 2 above, it can be observed that 5-coordinated Ge atoms correspond to 16.3% of the coordination environments with respect to the local geometry of the Ge atoms in the glass models with mid-gap states. Hence, by translating this into absolute numbers, in a typical glass sample with a defect state in the band gap, 11 Ge atoms will have 5-coordinated local environments, taking also into account that the total number of Ge atoms in a glass model is 70.

We also made a SOAP map in which the distribution of points from the different amorphous models is compared to crystalline-like environments, shown in Fig. 4 below. Local

environments from crystalline configurations, as well as those from the fragments in which the identified mid-gap states are localized within the amorphous network of the modelled systems, are found on the SOAP map to be towards the edges of each species cluster, and, in particular, in regions corresponding to over-coordinated atoms. This observation from this configurational map conveys our view that 5-coordinated Ge atoms correspond to a local environment that can be found inside the glass models and it is able to participate in crystalline-like atomic structural motifs, which, in turn, can host the mid-gap electronic states of the amorphous material.

Something like that does not imply that all these 5-coordinated Ge local environments will result, *a priori*, in a crystalline-like structural pattern, which was found to be responsible for the localization of the defect, since this structural motif appears to be more complex than that and we identified that 4-fold connected rings, for example, are also present in the atomic geometry of the defect, contributing to the overall picture of the environment that hosts the mid-gap electronic states inside the glass. Based on the percentage of the glass models (35%) that have mid-gap electronic states in their band gap and in conjunction with the size of the modelled system (315 atoms), we expect to find, on average, one such crystalline-like environment in every 900 atoms. We would like to highlight that this estimate from the statistics from the database of the small models was verified after our calculations (GAP-MD simulations + hybrid-DFT geometry optimization) in a 900-atom 225GST glass structure. The mid-gap defect electronic state present in the band gap of this larger model is also hosted by a very similar crystalline-like atomic environment within the amorphous network (see our reply to Referee #2).

Revisions: We added Fig.4 in the Supplementary Information, together with some relevant discussion. We commented in the main paper about the “frequency” that a crystalline-like environment similar to the one observed to be responsible for the localization of the mid-gap defect can be found inside the glass structure.

Fig. 4. Configurational SOAP map of the Ge, Sb, and Te local environments for each atomic species in the amorphous 225GST structures generated in this work, coloured by the nature of the structure in which the environment is found; partially crystalline (green empty circles), amorphous with mid-gap defect state (blue empty circles) and amorphous without in-gap electronic states (red circles).

Page 13 - paragraph 3.

The modified text reads: “Five-coordinated Ge atoms correspond to a defective local environment within the amorphous network, which by default means that it is not a dominant environment. A configurational SOAP map presented in **Fig. S38** in the Supplementary Information conveys the view that 5-coordinated Ge atoms are able to participate in crystalline-like atomic structural motifs, which, in turn, can host the mid-gap electronic states of the amorphous material. However, this does not imply that all these 5-coordinated Ge atoms will result, *a priori*, in a crystalline-like structural pattern, which was found to be responsible for the localization of the defect, since this structural motif appears to be more complex than that, and 4-fold connected rings, for example, were also identified in the atomic geometry of the defect, contributing to the overall picture of the environment that hosts the mid-gap electronic states inside the glass. Based on the percentage of the glass models ($\approx 35\%$) that have mid-gap electronic states in their band gap, and in conjunction with the size of the modelled system (315 atoms), we expect to find, on average, one such crystalline-like environment in every 900 atoms.”

Reviewer #1 Q3

Unoccupied gap-states are found in 7 samples of the 20 structures. The percentage of having the gap states is 35%, which is quite low. The authors mention that gap state is associated with crystalline-like environment having low local stress. Is this environment energetically stable? If so, why it is missing in most of the samples? Can the authors comment on this?

Authors: In Table 1, we report the total energy for each amorphous 225GST modelled system after the geometry relaxation of the glass models with hybrid-DFT calculations. In addition, Fig. 5 highlights the energy differences between glass samples with in-gap electronic states in the band gap and glass samples without in-gap states. It can be observed that the glass models with additional electronic states in their band gap have higher total energy, on average, than the 225GST samples with a “clean” band gap. By considering the relatively low amount of 5- and 6-coordinated Ge atoms in the amorphous structure, such environments can be associated with a lower Boltzmann weight and a higher potential energy (on average); therefore, they can be viewed as being unstable within the amorphous matrix. It is also worth mentioning that such defective octahedral coordination is relatively common in the liquid phase of 225GST, and therefore is not necessarily a hallmark of crystalline energetic stability in the absence of periodicity. Hence, this is an indication that the glass structures with in-gap states are unfavourable from an energetic point of view. Nevertheless, the average difference in energy between configurations with and without in-gap states is small (no more than 10meV/atom), and consequently the structural differences between the models are relatively subtle.

Table 1. Total energy of the relaxed ground state for the 30 glass models generated in this work. The energy of the relaxed 225GST glasses with electronic states in the band gap is, on average, higher than that of the amorphous systems with no additional electronic states in their band gap.

Glass model	In-gap states	Total energy (hartree)
1	NO	-2086.605
2	YES	-2086.523
3	NO	-2086.682
4	YES	-2086.616
5	YES	-2086.672
6	NO	-2086.694
7	NO	-2086.683
8	YES	-2086.581

9	YES	-2086.680
10	NO	-2086.651
11	NO	-2086.639
12	NO	-2086.708
13	YES	-2086.551
14	YES	-2086.680
15	NO	-2086.684
16	NO	-2086.623
17	NO	-2086.601
18	YES	-2086.618
19	NO	-2086.564
20	NO	-2086.677
21	NO	-2086.602
22	NO	-2086.671
23	NO	-2086.597
24	YES	-2086.745
25	NO	-2086.666
26	NO	-2086.688
27	NO	-2086.655
28	YES	-2086.668
29	NO	-2086.721
30	YES	-2086.678

Fig. 5. Comparison of the energetic character between the glass models with in-gap states (True) and without gap states (False) in the band gap. One can observe that the glass structures with in-gap states tend to be more unfavourable from an energetic perspective.

We understand the query from the Reviewer why in-gap states are missing from 65% of the glass samples. To answer this question, we would like to give some more perspective regarding the computational modelling of these defects. In a real glass sample of amorphous 225GST, one would expect to find in-gap states in the band gap of the material. However, the concentration of these electronic states is relatively low. The defect-state density within the band gap was measured by modulated photocurrent experiments [2] and it was found to be: $1 \times 10^{10} / \text{cm}^3 \text{ eV}$ for the mid-gap defect states and $4 \times 10^9 / \text{cm}^3 \text{ eV}$ for shallow gap states (located $\sim 0.2 \text{ eV}$ from the valence-band edge). Consequently, the in-gap states in a material like amorphous 225GST correspond to a difficult task to be modelled for systems of small size. Therefore, calculations in many models, of the size of ~ 300 atoms, are required in order to be able to trace the in-gap states present in the electronic structure of the amorphous material, taking also into account that hybrid functionals are mandatory for an accurate description of the electronic properties. By simulating many small models, we were able to explore more different structural environments from a statistical point of view, and hence identify successfully the in-gap states (mid-gap and shallow defects) in the band gap of the glass.

We also note that, after receiving the reports/comments from the Reviewers, we decided to generate 10 more glass models of the same system size, making the total number 30, to increase even more the statistical accuracy of our database of glass samples and the relevant conclusions for the in-gap electronic states (see our reply to Reviewer #3 Q5). In addition, we decided to push the boundaries of the system size by generating a 225GST glass model of 900 atoms and calculating its electronic structure, again with hybrid-DFT (see our revisions with respect to the reply to Reviewer #2). The fact that we found multiple defect states in the band gap of the 900-atom glass model not only verifies the accuracy of our database, but also highlights, with state-of-the-art computer simulations, the nature of the mid-gap electronic states expected to be present in a real sample of amorphous 225GST.

Revisions: We added Table 1 and Fig.5 in the Supplementary Information and we commented about the energetic character of the relaxed glass structures in the main paper. In the Introduction section of the paper, we also commented about the experimental measurements for the defect-state density in the band gap of the amorphous material.

Page 3 - end of paragraph 3.

The modified text reads: “The defect-state density in the band gap of glassy $\text{Ge}_2\text{Sb}_2\text{Te}_5$ has been measured by modulated photocurrent experiments²⁸ and was found to be: $1 \times 10^{10} / \text{cm}^3 \text{ eV}$ for the mid-gap defect states and $4 \times 10^9 / \text{cm}^3 \text{ eV}$ for shallow gap states (located at $\sim 0.2 \text{ eV}$ from the valence-band edge), highlighting that the concentration of these electronic states

is relatively low, which makes these defects difficult to be traced with computational modelling.”

Page 13 - paragraph 2.

The modified text reads: “It can be noted that the glass structures show also some subtle differences from an energetic point of view. The total energies of the thirty modelled amorphous 225GST systems after geometry relaxation with hybrid-DFT calculations are reported in **Table S2** in the Supplementary Information. In addition, **Fig. S37** highlights the energy differences between glass samples with in-gap electronic states and glass samples without in-gap states. It can be observed that the glass models with additional electronic states in their band gap have a higher total energy, on average, than the 225GST samples with a “clean” band gap. By considering the relatively low amount of 5- and 6-coordinated Ge atoms in the amorphous structure, such environments can be associated with a lower Boltzmann weight and a higher potential energy (on average); therefore, they can be viewed as being unstable within the amorphous matrix. Nevertheless, the average difference in energy between configurations with and without in-gap states is small, which is in accordance with the observed subtle structural differences between the models.”

Reviewer #1 Q4

More discussions are needed as the authors try to link the gap-states or 5-coordinated Ge atoms to the resistance-drift of the glass structure. It is unclear in the present manuscript.

Authors: We completely agree with the Reviewer that we need to establish a better connection between the in-gap states in the band gap and the resistance drift of the amorphous phase of 225GST. Hence, we have worked in the direction to improve our view regarding this aspect.

Revisions: We revised the Conclusions section of the paper in order to connect the observations from our simulations to the resistance drift of the amorphous phase and also to give some perspective on how the problem could be tackled in future PCRAM devices.

Pages 20 and 21.

“The physical origin of the resistance drift in phase-change memory devices, and its relation to the structure of the amorphous phase of phase-change memory materials, is a widely debated issue in the literature. It has been ascribed previously to the spontaneous structural relaxation (“ageing”) of the amorphous state created via the melt-and-quench

process^{27,66}. In experimental studies of glassy GeTe, the resistance drift of the amorphous phase was correlated with changes in the density of states upon structural relaxation^{2,18,19}. The decrease of the conductivity was explained by a shift of the in-gap defects and band edges away from the Fermi level, simply due to a stretching of the density of states, i.e. a larger energetic hopping distance, which also leads to a movement of the Fermi level nearer to mid-gap⁶⁷. First-principles calculations on amorphous GeTe indicated also a correlation between the resistance increase and the annihilation of structural defects responsible for localized electronic states, which results in an increase of the band gap of the glass and occurs via a series of collective rearrangements of the defective structural complexes^{11,66}.

Based on this perspective, the effect of ageing results in the amorphous phase evolving over time towards an energetically more favourable “ideal” glass state^{27,66}. Our calculations could support this view and provide some indication of the effect in amorphous 225GST, since the glass models among the thirty 225GST glass samples with in-gap electronic states were found to be energetically less favourable compared to the modelled systems which do not have defect states in their band gap (see Table S2 and Fig.S37 in the Supplementary Information). Hence, upon structural relaxation, one would expect the in-gap states to disappear gradually with time from the band gap of the glassy models, in order to reduce the overall free-energy of the system, thus resulting in more energetically stable amorphous systems and leading to an increase of the resistance, as the glass becomes more similar to some “ideal” amorphous configuration. However, we would like to note that the exact nature of such an “ideal” amorphous state of 225GST requires further investigation.

Nevertheless, a different view about the resistance drift could be drawn, which does not involve structural ageing effects, that is inspired by the unique way that amorphous 225GST is made in PCRAM devices. Instead of simply quenching a liquid formed by the external thermal melting of a crystal, as for traditional glass formation, the thermal energy for melting in a PCRAM cell is provided internally by the Joule heating associated with the application of a RESET voltage pulse. The *I-V* characteristics of 225GST are non-linear and the non-Ohmic increase in current is due to electric-field-assisted carrier-generation processes¹⁵. Hence, when the amorphous phase of 225GST forms on quenching from the liquid in a PCRAM cell, it is in the presence of a very high electron-hole current density. Thus, under such conditions, most of the localized gap-state trapping centres, characteristic of disordered materials and created during the formation of the amorphous phase, become filled during the RESET process itself¹³. Thereby, the electronic condition of the material immediately post-RESET is equivalent to that an amorphous semiconductor that can be found after steady-state pulsed optical excitation and before the following transient photocurrent decay. Electrons/holes are subsequently thermally de-trapped and an applied 'read' voltage produces a time-dependent current, which can be interpreted as a time-dependent resistance. Consequently, this type of

behaviour can be used to explain the very short-time increase in resistance evident in amorphous 225GST immediately after the reset pulse⁶⁸. It is noted that this approach has been previously used to describe the long-time photocurrent decay exhibited by amorphous silicon⁶⁹.

This alternative electronic mechanism for the origin of the resistance drift is based on the fact that the in-gap localized states in the band gap of the amorphous phase, present due to the structural disorder, can serve as charge-carrier traps, and the resistance-drift effect is related to the slow deep-trap release and subsequent recombination of such charge carriers⁶⁸. From our electron-trapping calculations in this study, we highlighted that the unoccupied mid-gap defect electronic states in the band gap of the amorphous 225GST models are able to capture electrons, leading to deep-trap states with strong electron localisation in the structural motif of the defect. This observation works in favour of the view that the resistance drift in amorphous 225GST might be caused by electron/hole trapping during the electron/hole-injection process accompanying the RESET process and creation of the glass.

It has been observed that the SET (crystalline) state of 225GST does not exhibit resistance drift^{12,17}, behaviour that can be understood due to the absence of localized in-gap states in the band gap of the metastable rock-salt crystalline phase of 225GST, which can act as slow-release trapping centres observed in the amorphous phase of the material. Hence, there is the prospect of mitigating resistance drift in phase-change memory devices by suitable material engineering to control the extent of band tailing and the distribution of the related localized in-gap defect states within the band gap of the amorphous phase, thereby facilitating the realization of multi-level storage operation in future PCRAM devices.”

Reviewer #1 Q5

In this work, glass structures obtained from MD simulations are further optimized using DFT with hybrid functional. It will be very meaningful for the reader if the authors can show the difference between the relaxed structures based on the hybrid functional and those based on GGA/LDA which is normally used in the stage of structure optimization.

Authors: We appreciate the comment and the respective suggestion from the Reviewer; however, we do not agree with the Reviewer’s view that a comparison between our relaxed glass structures with the hybrid functionals and the relaxed geometries with GGA/LDA calculations will be meaningful with respect to the objective of our work. A geometry relaxation within the GGA approximation is an approach that is usually employed to minimize the energy of the glass structure (obtained with molecular-dynamics simulations) with respect to the atomic coordinates of the system when one is just trying to focus on the structure of the glass

model, wanting also to give some DFT-oriented perspective in the analysis of the structural properties. In our work, the inclusion of the Hartree-Fock exchange in the geometry optimization of the modelled systems is crucial for the electronic-structure calculations in order to be able to describe correctly the band gap of the glass, and hence being able to identify the in-gap electronic states. In other words, relaxation of the structures with PBE, for instance, would be irrelevant, since such geometry optimization results in obtaining an electronic structure with *no* band gap and, consequently, an electronic density of states in which no in-gap states can be found. Hybrid-functional DFT calculations are essential to identify the localized defect states in the band gap of amorphous 225GST, which then can be ascribed to specific structural features within the amorphous network. Hence, we would like the reader to view our work and the simulations presented in the manuscript through this perspective.

We would like to note, though, that in a previous modelling study, it was reported that hybrid-DFT geometry optimizations result in liquid and amorphous structures of 225GST in better agreement with the experimentally reported structures, and in particular, for the Ge–Te and Sb–Te interatomic distances [3].

Revisions: Based on the objective of our work, we do not believe that we will be able to draw any constructive arguments and conclusions by performing such a comparison; therefore, we prefer not to develop this line of discussion at this point. Please see our reply and revisions related to Reviewer #3 Q2 about the effect of different DFT-schemes on the structure of glassy 225GST and the main reasons for the choice of a hybrid functional for the geometry relaxation of the amorphous models generated in this work.

Reviewer #1 Q6

In the Method section, the authors may describe the calculation procedure of the local stress in the amorphous structure with more details.

Authors: We agree with the Reviewer that it would be useful for the reader to provide some details regarding the calculation of the local stress in the amorphous structures. The stress tensor for each atomic species in the 315-atom 225GST model was calculated by using the GAP framework. The stress tensor (σ) corresponds to the derivative of the total energy with respect to scaling of the cell encoded in the strain (ϵ):

$$\sigma_{\alpha\beta} = -\frac{1}{\Omega} \frac{\partial E}{\partial \epsilon_{\alpha\beta}}$$

in which α and β are cartesian components. The GAP model provides a local decomposition of the total energy, as emerged from the local atomic environments, indexed here by i :

$$E = \sum_i E_i$$

Hence, in that case, by taking the derivatives with respect to the cell vectors, and scaling with the total volume, on both sides of the equation above, the total stress can be written as a sum of local atomic virial stresses:

$$\sigma = \sum_i -\frac{1}{\Omega} \frac{\partial E_i}{\partial \epsilon_{\alpha\beta}} = \sum_i \sigma_i$$

The trace of the stress tensor will give then a “local atomic pressure” for each atom in the modelled glass system, which corresponds to the property that was shown in Fig. 5 of the manuscript:

$$p_i = \frac{1}{3} \text{Tr}[\sigma_i]$$

We would like to highlight that this methodology is exploiting the fact that, in a local potential like GAP, the total energy can be written uniquely as a sum of local terms, and hence being more interpretable for each atom in the simulation cell [4], while, in contrast, the total energy from a DFT calculation cannot be written uniquely in a similar way. Nevertheless, the total value of the stress in the glass structure is comparable between DFT and GAP calculations, since the GAP potential was trained using stress values from the DFT data that correspond to the reference training set [5].

Revisions: We added a description for the calculation of the total stress in the glass structure in the Supplementary Information due to publication-limited space reasons. We commented in the main paper about the approach followed in the relative calculation presented in the manuscript.

Page 16 - paragraph 1.

The modified text reads: “We note that the approach followed to calculate the local pressure inside the glass structure is exploiting the fact that, in a local potential like GAP, the total energy can be written uniquely as a sum of local terms, and hence being more interpretable for each atom in the simulation cell⁶³. A more detailed description of the calculation of the local stress in the amorphous structures is presented in the Supplementary Information.”

Reply to Reviewer #2

Reviewer #2 Q1

The first reason is that the manuscript does not advance substantially the field, since it does not shed light on the resistance drift phenomenon. The authors do not propose convincing models to describe the drift. Are the crystalline-like atomic environments expected to disappear during the relaxation of the amorphous network? If yes, why? Should the disappearance of these environments lead to a decrease in energy?

Authors: We would like to thank the Reviewer for finding our work interesting and we appreciate his/her criticism related to our results. We urge the Reviewer to look into our answers for **Reviewer #1 Q3, Q4 and Q5** in order to see our analysis for the energetic character of the 225GST glass models with in-gap electronic states, the challenge with respect to the computational modelling of these defects, our additional discussions in the manuscript regarding the connection between our simulations for the defect states and the resistance-drift phenomenon, and why we want to focus on the analysis of the glasses using hybrid-DFT calculations than that for structures which result from GGA calculations with respect, always, to the objective of our work. Nevertheless, we would like to answer the comments of the Reviewer separately as well, in order to clarify our work and highlight some of our revisions, which were the outcome of his/her suggestions.

We are afraid that we do not agree with the Reviewer's view about the novelty of our work, since we believe that we have carried out very comprehensive, statistically significant and state-of-the-art simulations in order to link the electronic density of states to the structure of the glass and to establish the atomistic structural motifs which are responsible for the localized mid-gap electronic states in the band gap of the amorphous phase of 225GST. A few groups have been working experimentally and with simulations on the problem of ageing of the amorphous phase in phase-change materials and its connection to the defect states in the band gap of the glass. However, the majority of these studies concentrate only on the very much simpler case of the binary composition of GeTe, which is of little technological interest.

Our simulations correspond to the first atomistic modelling study of the in-gap defect states in amorphous 225GST, which is a ternary material system with extremely complex chemical bonding and is the prototypical material used in phase-change memory devices. There are some experimental studies which investigated before the defect concentration in the band gap of amorphous 225GST [2,6,7,8], but our study reveals, for the first time, the atomistic nature of these defects, which are undoubtedly present in the band gap of the amorphous material, and this was completely unknown before our calculations. In addition, the capability of the GAP potential employed in our calculations is also very significant, since it allowed us to

simulate many independent glass models, thereby giving us the opportunity for a proper statistical study, necessary for the investigation of the in-gap states in amorphous 225GST.

Revisions: Please see our revisions in the paper with respect to Reviewer #1 Q1, Q2, Q3 and Q4.

Reviewer #2 Q2

The second reason is that I have concerns about the validity of the findings reported in this work. My main concern is related to the accuracy of the GAP: in their previous paper (Ref. 29), the authors have shown that the GAP amorphous models have a smaller number of tetrahedral Ge (5%-11%) than the DFT models (20%-30%) (note that, in the present work, it is instead claimed that "more than 50% of the Ge atoms are tetrahedrally coordinated within the amorphous network". I assume that, by "tetrahedrally coordinated", it is meant here that the Ge atoms have 4 nearest neighbors; if not, there would be an alarming discrepancy between this work and Ref. 29). In Refs. 27 and 46, it was shown by DFT simulations and classical simulations with neural-network potentials, respectively, that tetrahedral Ge and wrong GeGe bonds (which stabilize tetrahedral Ge) play a crucial role in the ageing process of GeTe - the parent phase of GST. In these works, relaxation of amorphous GeTe was induced by chemical substitution and metadynamics, respectively: the relaxation process was shown to be driven by the disappearance of Ge-Ge chains and tetrahedral structures. Moreover, in Ref. 46, it was argued that similar chains of wrong bonds (Ge-Ge, Ge-Sb, Sb-Sb) are present in GST and disappear during ageing. Since the number of tetrahedra (and wrong bonds, I guess) in the GAP models deviates significantly from the DFT ones, I doubt that it is possible to extract useful information on the ageing process from the GAP models.

Authors: We would like to thank the Reviewer for pointing out the inconsistency in our statement related to the "tetrahedrally" coordinated Ge atoms inside our glass models. As the Reviewer correctly highlights, we really meant 4-coordinated Ge atoms, i.e. Ge with four other atoms as first-nearest neighbours in a defective octahedral, not just a tetrahedral, arrangement. We acknowledge our mistake and we corrected the statement accordingly wherever present in the manuscript. Nevertheless, we do not believe that the Reviewer had to go to such great lengths regarding the accuracy of the GAP potential. The structure of the glasses that were generated by using the potential were successfully validated against experimental data and previous DFT-modelling studies, highlighting that GAP models can be indeed representative of the glass.

The Reviewer also is concentrating on the inset of the Fig. 2d from our previous work [Ref.29 of the manuscript], which corresponds to the distribution of the tetrahedral angular order parameter (q_4), used to estimate the fraction of tetrahedral-like environments in the glass structures, and which shows a discrepancy between the GAP and the DFT models. Firstly, we would like to highlight that the proportion of tetrahedral-like environments in phase-change materials is a debatable subject and a question with no certain answer yet, while the q_4 order parameter is relatively sensitive to the simulation set-up. From Fig. 2d and the rest of the analysis for the local atomic structure, the take-away message is that, even though the GAP potential may slightly under-estimate the proportion of tetrahedral environments and the amount of homopolar bonds in the glass structure, it can capture the coexistence of tetrahedral and octahedral environments within the amorphous network of 225GST, as previously reported from *ab initio* MD simulations [9]. It is very important to realize that glass models generated with the GAP potential are not missing any of the atomic environments present in amorphous 225GST.

Moreover, we recommend the Reviewer to focus on Fig.4 of our previous work [Ref.29 of the manuscript], since this figure contains analysis and information directly relevant to the work presented in our current manuscript. This figure demonstrates that the GAP glass models are able to successfully reproduce the complex distribution of local environments present in amorphous 225GST. All the different structural motifs related to Ge coordination environments were identified inside the amorphous network of 225GST (trigonal pyramidal, tetrahedral, see-saw and square pyramidal). Finally, as we already mentioned in our current manuscript, the calculated Ge local environments from our previous twenty (now thirty) glass models agree very well with the proposed geometry of Ge atoms in amorphous 225GST from experimental studies [Ref. 23 and 51 of the manuscript], as well as with the Ge local geometries in the glass, as reported from *ab initio* MD simulation studies of amorphous 225GST [Ref. 24, 25, 26, 50 and 52 of the manuscript].

As the Reviewer correctly said, the studies that are mentioned in his/her comment [Ref. 27 and 46 of the manuscript] and, of course, cited in our manuscript, are related to the ageing process of GeTe, which is a material that is a parent phase of GST. However, this has a much simpler structure compared to the extremely complex chemical bonding of the ternary material system of Ge-Sb-Te. We trust the results presented in these previous studies in amorphous GeTe. However, we do not think that the Reviewer is making a valid comment by saying that in Ref.46 the authors “argued” that their findings in GeTe are “probably” the case also for GST. We do not think that one can judge the validity of our own work in GST based on “speculations” mentioned in other studies of different materials. We would like also to draw the attention of the Reviewer to a study from the same (more or less) authors (from the group of Prof Bernasconi) that was also cited in our manuscript and is Ref.26. This is a first-principles

simulation study of amorphous 225GST, and in this paper, the authors “argue” from their calculations in a single amorphous model (of ~250 atoms) that Sb–Te chains are responsible for the “localization” of the mid-gap defects in amorphous GST and not chains of wrong bonds. Since these mid-gap defects play an important role during the structural relaxation of the glass, we believe that one can spot the inconsistency between the “argument” the authors constructed for GST from their calculations in GeTe and the geometric character of “localization” for the mid-gap defects in GST, as mentioned in a different work.

To summarize, we do not want, by any means, to judge the work of other authors, since we believe that all of the previous work available in the literature was extremely useful for us to conduct our own study. However, we doubt that one can just solely judge our work based on arguments made in other studies for different materials, which are also conflicting for the case of GST.

Revisions: We corrected our false statement throughout the paper. We also commented in the paper about the performance of the GAP potential and what it can do well regarding the structure of amorphous 225GST and what it cannot do so well, respectively. Please see also our revisions in the paper with respect to Reviewer #3 Q3.

Page 5 - paragraph 2.

The modified text reads: “The GAP potential exhibits an accuracy close to that of the underlying DFT-PBEsol training set, as described in our previous work³⁰. In addition, we have demonstrated that the machine-learned GAP potential can be employed for rigorous modelling, with near-DFT accuracy, of the short- and medium-range-order structure of amorphous 225GST. Analysis of the local atomic structure showed that, even though the GAP potential may slightly under-estimate the proportion of tetrahedral Ge environments and the amount of homopolar bonds in the glass structure, it can capture the coexistence of tetrahedral and octahedral Ge environments within the amorphous network of 225GST, as previously reported from *ab initio* molecular-dynamics simulations²⁴. Consequently, the complex distribution of local environments that coexist inside the glass can be successfully reproduced, and hence glass models generated with the GAP potential are not missing any of the atomic environments believed to be present in amorphous 225GST³⁰.”

Reviewer #2 Q3

I also wonder why the authors have investigated such small system size (~300 atoms). Since the authors employ classical GAPs, they could have afforded to study significantly larger models (an order of magnitude larger, I guess). I understand that they also performed DFT simulations with hybrid functionals afterwards, however I don't think this is a good reason not to generate larger models. Even if hybrid DFT calculations are not feasible, it is still very interesting to analyze the structural properties of large GAP models and determine the occurrence of different structural motifs to improve the statistics. Did the authors compare the hybrid-functional calculations with standard GGA simulations? The latter functionals can yield spurious charge delocalization (and definitely underestimate the gap), but, maybe, they are good enough to grasp the localization of the states hosted by the crystalline-like atomic environments. If this is the case, one could combine the generation of large-scale GAP models with GGA simulations of the electronic properties.

Authors: We do not agree with the argument from the Reviewer that, since we employed classical MD simulations, the system size of our glass models is small and there is a need to study larger structures. It is not a question of how large a model can be that one can afford to generate with classical MD melt-and-quench simulations, since it is common knowledge that not just an order of magnitude, but even larger models can be generated with classical MD simulations. However, we would like to point out, at the outset, that using spin-polarized DFT with nonlocal functionals for significantly larger periodic cells and the large number of structures required for obtaining statistics is computationally prohibitively expensive. Hence, a large model (more than 1000 atoms) would not be practically of any use, since it is computationally impossible to perform hybrid-DFT calculations for its electronic structure.

We would like to clarify the computational philosophy behind the methodology employed in our work, which is significant for the study of defects in an amorphous material. We believe that it is more important to generate many glass models and calculate their electronic structure with hybrid-DFT simulations than generating a single (large) glass structure and to calculate its electronic structure with standard GGA/LDA calculations. There are numerous studies in the literature that have employed this computational approach to investigate the defects in several amorphous materials, like amorphous silica [10,11,12], sodium silicate glasses [13], amorphous alumina [14,15], amorphous hafnia [16,17] and glassy GeTe [18]. In all of these studies, many independent glass models, with system sizes ranging between 200 and 400 atoms, were generated with classical MD simulations (by using several interatomic potentials depending on the availability for the system under study), and then hybrid-DFT geometry optimizations were employed for the calculation of the electronic structure. The reason that one needs to follow this computational approach is that it is important to apply the nonlocal

functional calculations to identify and characterize the localized defects in the band gap of the amorphous material.

Hence, we would like to elucidate how instrumental, crucial, and of utmost importance it is to perform electronic-structure calculations with hybrid functionals, since standard GGA/LDA simulations will not just under-estimate the band gap but they will result in NO band gap for any size of glass model of amorphous GST. Such a situation means that we will not be able to identify any localized in-gap states, which is the main objective of our present work. Therefore, we need to employ hybrid-DFT calculations to identify the defect states in the band gap of glassy 225GST, which then can be ascribed to specific structural motifs within the amorphous network based on the localization of the wavefunction for the molecular orbital.

In order to prove this to the Reviewer, we generated an amorphous 225GST model of 2520 atoms following the same melt-and-quench protocol used for the 315-atom glass structures. The geometry of the final quenched configuration at 0K was optimized within the GGA approximation by using the PBE functional to calculate the electronic structure of the glass model. The total electronic density of states of this model is presented in Fig. 6, in which it can be observed that there is no band gap for the glass model, which is unphysical, since amorphous 225GST is a semiconductor which has a certain value for its band gap (≈ 0.7 eV). The outcome of no band gap results also in the absence of any localized defect states. Consequently, the approach to combine a large-scale glass model with GGA calculations of the electronic properties is not sufficient, since it does not lead to the scientific objective that was the motivation of our current work presented in this manuscript. Therefore, we strongly believe that it is purposeless to explore very large models using GGA calculations, since we will not gain any important information for their electronic structure and the in-gap defect states, which are significant with respect to the objective of our work.

We would like to point out that the cell size of the modelled systems (~ 300 atoms) was chosen on the basis of several other studies of defects in amorphous materials mentioned above, while the number of the glass samples (30 models) was determined by the computer resources available for spin-polarized DFT calculations using nonlocal functionals. The value of the band gap of glassy 225GST is reproduced very well in our simulated systems, while our hybrid-DFT calculations capture the mid-gap defect electronic states present in the real material; therefore, we are confident that the obtained statistics are sufficient for the conclusions reached in our current study.

Fig. 6. Total electronic density of states (DOS), highlighted with a black solid line, and the corresponding inverse participation ratio (IPR) values for the Kohn-Sham orbitals, highlighted with red spikes, near the Fermi level for an amorphous 225GST model of 2520 atoms. The geometry optimization of the glass structure with a GGA calculation (by using the PBE functional) for the electronic structure results in no band gap for the amorphous model, which makes it impossible to identify any kind of defect electronic states.

Nevertheless, we would really like to thank the Reviewer for motivating us to try to perform simulations for a larger glass system size. By staying consistent within our computational methodology, we generated an amorphous 225GST model of 900 atoms with classical GAP-MD melt-and-quench simulations and then we calculated its electronic structure with hybrid-functional-DFT geometry optimization. We would like to mention that, even though this simulation was computationally very expensive, we are really pleased that we decided to do it. The electronic density of states for the 900-atom amorphous model, shown in Fig. 7, highlights the existence of several, deep (mid-gap) and shallow, localized unoccupied electronic states in the band gap of the glass. The visualization of the molecular orbital for the mid-gap electronic state, shown in Fig. 8, reveals also a very good agreement with what we had observed from the 315-atom glass models with respect to the atomic geometry responsible for the localization of the mid-gap electronic states within the amorphous network. It can be observed that this defect state is hosted by a crystalline-like atomic environment, while the involvement of 5-coordinated Ge atoms can be identified in the defect localization.

Fig. 7. Total electronic density of states (DOS), highlighted with a black solid line, near the top of the valence band and the bottom of the conduction band of an amorphous 225GST model of 900 atoms. Hybrid-DFT calculations were performed for the geometry optimization of the glass structure. The electronic-structure calculation shows a Kohn-Sham band gap of 0.63 eV for the relaxed ground state. Several, deep (mid-gap) and shallow, localized unoccupied electronic states can be identified in the band gap of the glassy model. The mid-gap electronic state is located at 0.37 eV below the conduction-band minimum. The corresponding inverse participation ratio (IPR) values for the Kohn-Sham orbitals, highlighted with red spikes, show strong spatial localisation of the mid-gap and shallow defect states.

Fig. 8. Atomic structure and molecular orbital of the mid-gap electronic state identified in the 900-atom amorphous 225GST model. 5- and 6-coordinated Ge atoms correspond to local environments that are involved in the localization of the mid-gap defect state, while the formation of 4-fold connected rings leads also to the creation of a cubic structural pattern in the geometry of the defect. Ge atoms are blue, Sb are red, and Te are yellow. The light green and blue isosurfaces depict the molecular-orbital wavefunction amplitude of the mid-gap defect state and are plotted with an isovalue of +0.015 and -0.015, respectively.

The electronic-structure calculation from this glass model verifies the statistical accuracy of the database of the previously 20 (now 30) 315-atom 225GST models, since from them we were also able to capture the mid-gap defects and their corresponding atomic environment inside the glass. We also believe that the hybrid-DFT electronic-structure calculation of the 900-atom model is novel from a computational point of view, since we really pushed the boundaries of the quality of electronic-structure calculations by performing such a simulation. To our knowledge, there is no previous study in the literature that has ever reported hybrid-DFT geometry optimization for such a system size of an amorphous structure. Hence, we think that the overall quality and novelty of our work has been massively increased.

Revisions: We added Fig.7 and Fig.8 from above in the main paper, together with a combined caption (see **page 14**) and some relevant discussion. We modified the text accordingly throughout the paper to reflect the calculations for the 900-atom glass structure. In addition, please see also our revisions in the paper with respect to Reviewer #1 Q5 and Reviewer #3 Q2.

Page 14 - paragraph 2.

The modified text reads: “This estimate from the statistics for the database of the thirty 315-atom glass models was verified after corresponding hybrid-DFT calculations for a 900-atom 225GST amorphous model. The total electronic density of states of the 900-atom glass structure, together with the calculated IPR spectrum near the band edges, shown in **Fig.5(a)**, highlights the existence of several, deep (mid-gap) and shallow, localized unoccupied electronic states in the band gap of the glass. The electronic-structure calculations show a band gap of 0.63 eV for the relaxed ground state of this larger amorphous model, while the mid-gap electronic state is located at 0.37 eV below the bottom of the conduction band. The visualization of the molecular orbital for the mid-gap defect state present in this modelled system, shown in **Fig.5(b)**, reveals an atomic geometry, responsible for the localization of the defect within the amorphous network, with a close resemblance to the one that hosts the mid-gap electronic states in the 315-atom series of glass models. It can be observed that a crystalline-like atomic environment is hosting again the mid-gap defect state, while the involvement of 5-coordinated Ge atoms can be identified in the defect localization.”

Page 15 - paragraph 1.

The modified text reads: “The electronic-structure calculation of the 900-atom glass model verifies the statistical accuracy of the database of the thirty 315-atom 225GST amorphous models, since from them the mid-gap defects and their corresponding atomic environment were also traced inside the glass structure. Moreover, the hybrid-DFT electronic-structure calculation for the 900-atom modelled system is novel from a computational point of view, since, to our knowledge, there is no previous study in the literature that has ever reported geometry optimization for such a large system size of an amorphous structure with nonlocal functionals.”

Reviewer #2 Q4

Furthermore, I do not understand why the quenching rates are so high (10^{13} K/s). The use of machine-learning potentials should also enable much longer simulations. Studying large models on long time scales should be the main reason for the development of a classical potential! Assessing the effects of quenching rates on the structure of amorphous GST is important. What happens if quenching rates of order 10^{12} K/s are employed? Would the systems crystallize during quenching? If yes, is this behaviour due to finite-size effects or to the GAP potential? Is it possible that the crystalline-like atomic environments are crystalline precursors, which reflect the extreme tendency of the system to crystallize?

Authors: We appreciate the suggestion from the Reviewer to investigate different cooling rates for the quench of the amorphous models, but we would like to clarify that this is beyond the scope of our current study. We would like to highlight that it would be undesirable to use a cooling rate that will result in crystallization of the modelled system [19], since we want to investigate the defects in the band gap of the amorphous phase of 225GST, while we note that a quenching rate of -15 K/ps that was employed in our simulations is a typical cooling rate applied for the generation of amorphous phase-change materials with molecular-dynamics simulations. We agree with the Reviewer that, by studying large models on long-time scales, one would be able to explore system-size and cooling-rate effects on the structure of the amorphous 225GST. Nevertheless, this clearly corresponds to a comprehensive study which is separate from the calculations we performed with respect to our current work. Hence, these questions are relevant to be addressed in a study with different objectives, which is beyond the scope of our present work.

Revisions: Based on the objective of our study, we do not believe that we will be able to draw any constructive conclusions by performing such simulations; therefore, we prefer not to develop this line of discussion at this point, since it is beyond the scope of our present work.

Reply to Reviewer #3

Reviewer #3 Q1

The machine-learned potential employed for generating multiple GST glass structures is claimed to produce near-DFT accuracy GST models with “excellent” agreement with experimental X-ray structure factors. However, in Fig. S2 only a qualitative sim. vs exp. comparison it is shown with only a direct comparison for the $g(r)$ DFT vs GAP (Fig. S1). A quantitative assessment of the structure factor comparison sim. vs exp. it would be definitely helpful to sustain the above claims. For instance, a straightforward way could be to use the goodness-of-fit parameter of Wright *et. al.* [J. Non-Cryst.Solids 71, 295 (1985)], which is widely employed in the glass community exactly for this purpose.

Authors: We agree with the Reviewer that a more quantitative comparison between experiment, DFT data and GAP-MD glass structures would be useful for the characterization of our amorphous models. However, we do not share the belief that use of the goodness-of-fit parameter used by Wright *et al.* in their study is a common practice employed in the literature related to computer simulations of glassy materials. To our knowledge, there are a great number of simulation studies in amorphous materials, in which glasses were generated with melt-and-quench molecular-dynamics simulations (classical or *ab initio*) and the comparison between experimental and modelling data for the radial distribution function and/or structure factor was implemented by plotting them on the top of each other [9,20-28 to mention a few]. To our mind, a criterion related to the goodness of fit would be useful in the case of Reverse-Monte-Carlo modelling for instance, in which experimental diffraction-pattern data are used as the basis to generate an amorphous model, and in such a case, the obtained glass structure is fitted to the experimental data.

Nevertheless, as we already mentioned, we agree with the Reviewer, and for this reason, we decided to compare the peak positions in the $g(r)$ and $S(q)$ distributions from Fig. S1 and Fig. S2, respectively, in the Supporting Information, in order to evaluate the quality of the modelled glass systems. We believe that such an assessment can serve adequately to provide a more quantitative perspective in a comparison between the experimental and modelling data, as well as between the GAP-MD glass structures and the DFT-MD data for amorphous 225GST.

The first, second and third nearest-neighbour distances, corresponding to the first three peaks in the radial distribution function in Fig. S1, are 2.89 Å, 4.09 Å and 6.23 Å, respectively, for the 225GST glass models generated by using the GAP potential, and 2.89 Å, 4.15 Å and 6.29 Å, respectively, for the amorphous 225GST model generated by *ab initio* DFT molecular-dynamics simulation. This comparison highlights the very good quantitative agreement

between the local atomic structure of the GAP-MD glasses and the DFT-MD amorphous model. Beyond these peaks, it can be observed that the agreement is less good between the glass models, since the GAP-MD samples do not capture correctly the fourth and fifth peaks in the $g(r)$ of the DFT structure. However, this discrepancy is not too significant, since the local atomic structure of the glass is mainly described within the first and second coordination shell, which are also important with respect to our observations for the localization of the mid-gap defect states within the amorphous network.

The first peak in the total X-ray structure factor is located at 2.07 \AA^{-1} in the experimental $S(q)$ distribution, at 2.11 \AA^{-1} in the $S(q)$ from the amorphous 225GST model generated by *ab initio* DFT molecular-dynamics simulation, and at 2.14 \AA^{-1} in the $S(q)$ calculated from the GAP-MD 225GST glasses generated in this work. In addition, the second and third peaks are located at 3.37 \AA^{-1} and 5.16 \AA^{-1} , 3.28 \AA^{-1} and 4.97 \AA^{-1} , 3.27 \AA^{-1} and 4.94 \AA^{-1} , in the same $S(q)$ distributions, respectively. Overall, it can be observed that the quantitative comparison between the peak positions in the relative $S(q)$ distributions indicates very good agreement between the experimental data and the amorphous 225GST models generated in our current study, with respect to the local atomic structure of the GAP-MD glasses.

Revisions: We added the discussion related to the quantitative assessment in the SI and we refer the readers of the paper to look at the Supplementary Information for a quantitative comparison between the data presented in Fig. S1 and Fig. S2 for the $g(r)$ and $S(q)$ distributions, respectively.

Page 6 paragraph 1.

The modified text reads: “A quantitative assessment between the experimental and modelling data, as well as between the GAP-MD glass models and the DFT-MD data for amorphous 225GST, also presented in the Supplementary Information, further highlights the very good agreement with respect to the first and second coordination shells of the glass structures.”

Reviewer #3 Q2

Moreover, more arguments should also be detailed on the specific choice of the DFT procedure used. In fact, multiple GST glass models have been developed in the last decade by first-principles molecular dynamics (FPMD) with different DFT schemes in terms of xc functional, pseudopotential, etc [Sci. Rep. 6, 25981 (2016); Phys. Rev. Lett. 103, 195502 (2009)] producing in certain cases even better quantitative agreement with exp. data [Phys. Rev. B 96, 224204 (2017); J. Appl. Phys. 113, 134302 (2013)].

Authors: We understand the comment from the Reviewer, since there are several studies in the literature about the structure of amorphous 225GST reporting the effect of the different DFT-schemes on the quality of the generated models. Therefore, we would like to give an overview of these approaches with respect also to the objective of our current work.

Electronic-structure calculations within the GGA approximation by using the PBE functional is an approach that satisfies a large number of exact constraints for the exchange and correlation functional. It is relatively inexpensive and non-empirical, while it corresponds to an approach that has been used with appreciable success in describing the atomistic structure of many amorphous solids, hence making it the “standard” choice for DFT calculations. However, it has been observed that PBE calculations result in longer Ge–Te and Sb–Te bond lengths compared to the experimental local atomic structures in amorphous 225GST, revealing, small, but significant discrepancies [3].

In Ref.29, mentioned by the Reviewer, the authors used the PBE functional to perform DFT calculations on 27- and 45-atom structures, while in Ref.30, the authors used the PW91 functional to explore the structure of 63-, 126- and 189-atom models. We would like to clarify that the size of the investigated modelled systems in these papers is very small and definitely of worse quality than that one can achieve nowadays with first-principles atomistic simulations, taking also into account the increase of the computational power available to the scientific community. Therefore, even though these two studies contain some useful information related to the chemical bonding inside the glass structure, we do not believe that they can be compared with our generated glass models.

The Reviewer also mentioned that in two other DFT studies, a better quantitative agreement was obtained for the structure of amorphous 225GST between experimental and modelling data [3,31]. We are aware of these studies and we think that both of them are very important papers in the field. However, we believe that such agreement is probably only superficially true, since amorphous structures of 144 atoms and 72 atoms were modelled in Ref.31 and Ref.3, respectively, which is indicative that both simulation studies might suffer from significant finite-size effects for the modelled system. In addition, it can be observed that, in Ref.3, the agreement in the structure factor between experiment and simulation is not that close beyond the first peak, indicating that not only the very small size of the modelled system (70 atoms), but also the very low plane-wave cut-off (250 eV for the liquid and 130 eV for the amorphous state) can cast some doubt on the convergence of energies, forces and stresses of the simulated structures. We would like to highlight, for comparison purposes, that in our calculations, a plane-wave cut-off larger by an order of magnitude (400 Rydberg \approx 5440 eV) was employed in the geometry optimizations of the glass models.

Moreover, Bouzid *et al.* claim in Ref.31 that the use of an alternative pseudopotential construction can provide an excellent calculation, even for a small model, for the structure

factor of amorphous 225GST. The authors demonstrated with their calculations that Troullier-Martins pseudopotentials perform better than GTH pseudopotentials. However, the fact that the authors lowered the ℓ value used for the spherical-harmonic basis of the pseudopotential construction (Kleinman-Bylander), in order to reach a better quantitative agreement with the experimental data, suggests that they may have benefited from fortunate-error cancellation, and that this agreement may not be transferable to other compositions or even other phases of GST.

Beyond the papers mentioned by the Reviewer, we would like also to point out that the PBEsol parametrization of the PBE functional can reduce the under-binding problem of the energies of the GGA approximation [32], while also it can provide a slightly better description of the amorphous structure of 225GST, without sacrificing the exact conditions that make PBE transferable [33]. The PBEsol functional, for instance, was employed to generate the amorphous structures that correspond to the training set used for the development of the GAP potential for amorphous 225GST [5]. Finally, it is worth noting that Van der Waals corrections may play an important role for the structure of this class of materials. Nevertheless, we opted not to explore something like that at this stage in order not to complicate even more our current, state-of-the-art, simulations.

In the study presented in Ref.3, the authors demonstrated that a geometry optimization with hybrid-DFT calculations decreases the estimated Ge–Te and Sb–Te interatomic distances in the simulated amorphous 225GST, leading to a better agreement with the experimental structure compared to the GGA geometry relaxations. This observation, we believe, validates our choice of using a nonlocal functional for the geometry optimization of the generated GAP-MD glass models (which already had a near-GGA accuracy).

In addition, regarding the calculation of the electronic properties, it is very well known that GGA calculations of the electronic structure under-estimate severely the band gap and the localized states of a semiconductor. Therefore, the inclusion of a portion of the Hartree-Fock exchange to the PBE approximation is imperative in order to be able to improve significantly the estimation of the band gap, and hence establish a better description of the electronic structure, which is crucial for the investigation of the localized mid-gap electronic states in amorphous 225GST.

Our study involves a two-step approach of generating GAP-MD amorphous models and then calculating their electronic structure with hybrid-DFT geometry optimizations. PBE0 was the chosen hybrid functional for our calculations, which is implemented efficiently within the CP2K code using the truncated Coulomb operator together with the GTH pseudopotentials [34], while also it offers more consistency with the GAP glass models over other hybrid functionals. In general, performing hybrid-DFT calculations with the CP2K code for many glass models is very efficient, since the computational cost of nonlocal functional calculations can

be reduced using the auxiliary density matrix method (ADMM), mentioned in the manuscript. With this approach, the density is mapped onto a much sparser Gaussian basis set containing less diffuse and fewer primitive Gaussian functions than the one employed in the rest of the calculation. This allows the Hartree-Fock exchange terms, whose computational expense scales as the fourth power of the number of basis functions, to be calculated on a much smaller basis set for the rest of the calculation, which substantially reduces the computational time [35].

Overall, we would like to note that our approach followed in the calculations presented in this work has a very good combination of accuracy, efficiency and transferability with respect to the objective of our study in identifying the localized mid-gap defect electronic states present in the band gap of glassy 225GST and revealing the local atomic structure that hosts these defects within the amorphous network.

Revisions: We commented in the paper about the studies that report the effects of the different DFT-schemes on the structure of glassy 225GST. We also explained why it is important to perform DFT calculations with hybrid functionals with respect to the objective of our work. From the computational point of view, we added a note in the Supplementary Information describing why CP2K is an efficient code to perform heavy hybrid-DFT geometry optimizations due to the ADDM implementation.

Page 12 - paragraph 3.

The modified text reads: "It is noted that there are several studies in the literature about the structure of amorphous 225GST reporting the effects of different DFT-schemes, with respect to the exchange-correlation functional^{56,57} and the pseudopotential⁵⁸ used within the generalized-gradient approximation (GGA), on the quality of the generated models. It has been observed that GGA calculations with the PBE functional result in longer Ge–Te and Sb–Te bond lengths compared to the experimental local atomic structures in glassy 225GST, revealing, small, but significant discrepancies⁵⁹. A geometry optimization with hybrid-DFT calculations decreased the estimated Ge–Te and Sb–Te interatomic distances in the simulated amorphous 225GST, leading to better agreement with the experimental structure compared to the GGA geometry relaxations⁵⁹. This observation validates our choice of using a nonlocal functional for the geometry optimization of the generated GAP-MD glass models, which already had a near-GGA accuracy. Moreover, the inclusion of a portion of the Hartree-Fock exchange to the PBE approximation is imperative in order to be able to improve significantly the estimation of the band gap, and hence establish a better description of the

electronic structure, which is crucial for the investigation of the mid-gap electronic states in amorphous 225GST. Hybrid-functional DFT calculations are essential to identify the localized defect states in the band gap of glassy 225GST, which then can be ascribed to specific structural features within the amorphous network.”

Reviewer #3 Q3

If I correctly understand the local (defective) coordination scenario of the GST species is analyzed using a geometrical bond-distance cut-off: this approach is arguably appropriate for such a complex type of chalcogenide glass. In fact, for GST and other chalcogenides the local coordination analysis has to be combined whether with Electron-Localization Function ELF (Adv. Mater., 29, 1700814 (2017)) or the Wannier centers formalism analyses (Phys. Rev. B 93, 115201 (2016); Phys. Rev. B 96, 224204 (2017)) to confirm that there is actual bonding interaction between the species. It would be instrumental for the reader to see if actually the mentioned defective 5-coordinate Ge atoms show any sort of interaction with all the nominally coordinated surrounding neighbors. And if yes, what the type of interaction they show.

Authors: We appreciate the constructive criticism of the Reviewer related to our structural analysis for the local coordination environments inside the glass models. We agree with the Reviewer that a more DFT-based analysis has been employed before in the literature to investigate the bonding between the atomic species in phase-change materials. A structural analysis by using the Electron-Localization Function (ELF) and/or the Maximally Localized Wannier Functions (MLWF) can provide a better view with respect to the chemical bonding and the local environments within the amorphous network.

Following the suggestion from the Reviewer, we calculated the local coordination environments for all the atomic species in our glass samples, based on information from the charge-density distribution and by using the ELF [36,37]. The coordination numbers of the three atomic species in the simulated amorphous 225GST structures, as obtained from the ELF analysis, are shown in Fig. 9. It can be observed that 3- or 4-fold coordination is the favourable arrangement for Ge local environments, while 3-fold coordinated environments are the most probable for Sb and Te atoms.

Fig. 9. Coordination-number histograms obtained from an Electron-Localization Function (ELF) analysis for the local environments of Ge, Sb, and Te atoms, averaged over the thirty 225GST glass models.

The average coordination numbers around Ge, Sb, and Te atoms, calculated from the thirty amorphous 225GST models with the ELF analysis, were found to be 3.7, 3.3, and 2.7, respectively, while the average coordination numbers around Ge, Sb, and Te atoms, calculated from the thirty glass samples by using a geometrical bond-distance cut-off of 3.2 Å, were found to be 4.2, 3.6, and 2.9, respectively. It can be observed that the ELF analysis results in smaller average coordination numbers for the three atomic species in the glass structures, while the geometrical bond-distance cut-off approach shows slightly over-structured local environments for the glass models. However, the differences are not very significant, since both approaches highlight, for instance, the large range of Ge local environments inside the glass structure. The amount of 5- and 6-coordinated Ge atoms was found to be 18% with the approach employed in the structural analysis presented in the manuscript, while with the ELF analysis, 11% of the Ge atoms were found to have 5 or 6 atoms in their nearest surroundings. More importantly, we would like to point out that, even though the geometrical bond-distance cut-off approach over-estimates the amount of 5-coordinated Ge atoms, the ELF analysis showed that this Ge local geometry is evidently present in the structural motif responsible for the localization of the mid-gap defect within the amorphous network of the glass samples, which gives us extra confidence with respect to our view for the local atomic structure of the mid-gap defect and the crucial role of the 5-coordinated Ge atoms.

We would also like to note that an optimal geometric bond-distance cut-off, that minimizes the number of errors, can be deduced from an electronic-structure calculation in several different ways [38,39]. A value for this cut-off between 3.0 and 3.2 Å is typically obtained for amorphous 225GST, which highlights that the radial cut-off distance of 3.2 Å used in our work for the structural analysis corresponds to a reasonable choice for describing the chemical bonds in 225GST in a correct manner. Overall, we believe that the geometrical bond-distance cut-off approach, combined with the SOAP structural descriptors for the local environments of the atomic species in the glass samples generated in our work, correspond to a very good description of the local atomic structure of the amorphous models.

Revisions: We added the ELF structural analysis for the local environments of Ge, Sb, and Te atoms in the Supplementary Information, together with a comment for the choice of the value for the geometric bond-distance cut-off. We compared the coordination environments obtained with the two different methods in the main paper, together with some comments related to the geometry of the mid-gap defect in the glass structure.

Page 10 - paragraph 3.

The modified text reads: “The average coordination numbers around Ge, Sb, and Te atoms, calculated from the thirty amorphous 225GST models by using a geometrical bond-distance cut-off of 3.2 Å, were found to be 4.2, 3.6, and 2.9, respectively. These results are similar to those reported by *ab initio* molecular-dynamics simulations of amorphous 225GST^{26,29,52}, as well as being in very good agreement with our recent work³⁰. The local coordination environments for all the atomic species in our modelled systems were also calculated, based on information from the charge-density distribution and by using the Electron-Localization Function (ELF)⁵³. The coordination-number histograms obtained from the ELF analysis for the local environments of the three atomic species in the glass models are shown in **Fig. S34** in the Supplementary Information, while the average coordination numbers around Ge, Sb, and Te atoms, calculated from the thirty glass samples with the ELF analysis, were found to be 3.7, 3.3, and 2.7, respectively.”

Page 12 - paragraph 1.

The modified text reads: “The ELF analysis results in slightly smaller average coordination numbers for the three atomic species in the glass structures, while the geometrical bond-distance cut-off approach shows slightly over-structured local environments for the glass models. However, the differences are not very significant, since both approaches highlight, for instance, the large range of Ge local environments inside the glass structure (see Supplementary Information for details). More importantly, it is worth highlighting that, even though the geometrical bond-distance cut-off approach somewhat over-estimates the amount of 5-coordinated Ge atoms, the ELF analysis showed that this Ge local geometry is evidently present in the structural motif responsible for the localization of the mid-gap defects within the amorphous network of the glass samples, which verifies the crucial role of the 5-coordinated Ge atoms in the local atomic structure of the mid-gap defects.”

Reviewer #3 Q4

Regarding the stress tensor calculations, it would be instrumental to know the exact value of the final glass models residual stress. Generally, a value of cell residual stress >1 GPa is an index of the fact that the glass model has not been relaxed sufficiently. Or, in another words, that the simulation is not predicting the density correctly.

Authors: The Reviewer brings up an important point related to the computational modelling of amorphous materials and the advantages of fixed volume (NVT) vs. fixed pressure (NPT) simulations. We would like to highlight that the main limitation of the NPT approach is that the

volume (and density) of the simulated system may not correspond to the optimal volume (and density) for the selected type of the interactions, whereas performing simulations with the NVT ensemble allows one to obtain structures fitting a pre-defined (in our case experimentally determined) density. Hence, in the simulations presented in this work, the density of the glass models was fixed to the experimentally reported value for amorphous 225GST.

In general, it is noted that fixing the simulation cell volume to match the target-glass density throughout the melt-and-quench MD simulations is a robust approach, employed and well tested in prior modelling studies in order to generate glass models representative of the experimental structures [13,18,20,21,23,33,40-49]. It also avoids complications due to the volume fluctuation, and thus reduces the equilibration time. However, the fixed-volume approach for the melt-and-quench process will generate a glass structure with relatively high final pressure and stress in the cell. The relaxation of the cell of the final glass structure with DFT calculations would have an effect on the density of the simulated glass but the difference is usually comparable to the DFT error [50]. Consequently, we believe that this residual stress, present in our glass structures, does not affect the quality of the calculations, since the effect of the cell relaxation is expected to be small on the volume of the amorphous models.

Hence, weighing these considerations, we opted for the NVT ensemble and geometry optimizations of the glass structures. We would like to clarify, though, that from the simulations presented in our work, we did not aim to predict the density of the glass; therefore, we fixed the density to the experimental value in order to generate 225GST glass models with good representative atomic structures, which is important with respect to the objective of our study in finding the structural motifs within the amorphous network that are hosting the mid-gap defect electronic states of the glass.

Revisions: We provided the values for the residual stress in the final glass models, as per the Reviewer's request, and we also commented about these values with respect to the computational methodology employed in our work for the generation of the amorphous models. Please see also our revisions related to Reviewer #1 Q6.

Page 17 - paragraph 1.

The modified text reads: "It is worth highlighting that, from the simulations presented in this work, we did not aim to predict the density of the glass; therefore, the density of the glass samples was fixed to the experimentally reported value for amorphous 225GST. It is noted that fixing the computational cell volume to match the target-glass density throughout the melt-and-quench MD simulations is a robust approach that can be employed to generate glass models with good representative atomic structures, which is important in finding the structural

motifs within the amorphous network that are hosting the mid-gap defect electronic states of the glass. However, the fixed-volume approach will result in a glass structure with residual stress in the cell. In the thirty 225GST amorphous models generated in this work, the total residual stress ranges between 1.3 and 1.7 GPa in the glass structures after the geometry optimizations with the hybrid-DFT calculations. We believe, though, that this does not affect the quality of the calculations, since the effect of the cell relaxation is expected to be small on the volume of the simulated systems⁶⁴.”

Reviewer #3 Q5

The strategy employed by the authors to investigate the origin of the resistance drift basically follows the scheme employed by Zipoli et al. [Phys. Rev. B 93, 115201 (2016)] with a detailed analysis of the mid-gap electronic states of multiple GST glass models. Although the results shown in this work and their interpretation sound to be comprehensive at this level, to rule out any doubt more statistics is needed. At this stage only #7 configurations have been used to draw the conclusions on the correlation between the mid-gap electronic states and specific local defective coordination environments of GST species. After all, one of the supposedly advantage of having developed for the first time a machine-learned potential for such complex chalcogenide glass system is the possibility to simulate larger systems or produce “good” GST glass models at less expensive computational cost than with purely FPMD/DFT schemes.

Authors: Following the suggestion from the Reviewer, we generated 10 more glass models with classical GAP-MD simulations, with the same melt-and-quench protocol, and then we calculated their electronic structure with hybrid-DFT geometry optimizations. Hence, our (small) database now contains 30 (thirty) glass structures of 315-atoms. From these samples, 11 models have in-gap states in their band gap, corresponding to ~37% of the total number of models. From the 11 modelled systems with in-gap states: a) 9 glass models have a well-defined mid-gap defect electronic state; b) 5 glass models have a shallow defect state (located ~0.2 eV from the valence-band edge); c) 1 glass model has two defect states; and d) 2 glass models have three defect states in their band gap.

As one can also see from our response to Reviewer #2, we decided to push the boundaries of the system size by generating an amorphous model of 900 atoms with GAP-MD melt-and-quench simulations and calculating its electronic structure, again with hybrid-DFT simulations. The observations from the electronic density of states, see Fig. 7, and the atomic environment related to the localization of the mid-gap defect of this model, see Fig. 8, complement our observations from the database of the smaller models.

Therefore, we believe that the statistical accuracy and the quality of our conclusions have been massively increased after the simulation of the 900-atom glass model, which, together

with the database of the 315-atom glass samples reveal, with state-of-the-art computer simulations, the atomistic nature of the mid-gap defect electronic states expected to be present in a real sample of amorphous 225GST.

Revisions: We revised the text accordingly in order to include the results from the calculations of the additional 10 glass models in the statistical analysis presented in the paper. We also updated Table S1 and its relevant caption in the Supplementary Information. We added the DOS of the 10 extra modelled systems in the Supplementary Information as well. Finally, we modified the text throughout the paper in order to reflect all the additional simulations performed after the revision process.

Page 6 - paragraphs 1,2,3.

The modified text reads: “The thirty glass structures obtained using the GAP interatomic potential were further optimised using DFT with a hybrid functional. The electronic-structure calculations show an average Kohn-Sham band gap of 0.66 eV for the relaxed ground state. **Table S1** in the Supplementary Information contains the values of the band gap for all the thirty glass samples. The calculated values of the band gap, which range from 0.55 to 0.79 eV, agree very well with the experimentally reported values for amorphous 225GST³⁸⁻⁴¹, ranging between 0.6 and 0.8 eV, as well as with previous modelling studies^{26,42}.

Of the 30 amorphous 225GST models studied here, in 11 glass structures, corresponding to $\approx 37\%$ of the glass samples, in-gap electronic states emerged in their band gaps. From the 11 modelled systems with in-gap states: 9 glass models have a well-defined mid-gap defect electronic state; 5 glass models have a shallow defect state; 1 glass model has two defect states; and 2 glass models have three defect states in their band gap. The mid-gap state is located, on average, 0.34 eV below the bottom of the conduction band and it corresponds to an unoccupied state in the band gap of the amorphous 225GST. The shallow unoccupied defect states are located, on average, 0.17 eV from the valence band-edge, in good agreement with experimental measurements. The positions of all the in-gap electronic states in the band gap of each glass model which has a defect state are shown in **Table S1**.

The total and partial electronic densities of states of a glass model (sample #4 from Table S1) with a mid-gap defect state is shown in **Fig.1**, whereas **Fig.S3** in the Supplementary Information shows the total and partial electronic densities of states of an amorphous 225GST model (sample #3 from Table S1) without the presence of any additional electronic states in the band gap, for comparison. For the sake of completion, the total electronic densities of states for all the thirty amorphous 225GST models generated in this work are shown in **Figs.S4 – S33** in the Supplementary Information.”

References

1. Bartók, A. P., Kondor R. & Csányi, G. *Phys. Rev. B* **87**, 184115 (2013).
2. Luckas, J., Krebs, D., Grothe, S., Klomfaß, J., Carius, R., Longeaud, C. & Wuttig, M. *J. Mater. Res.* **28**, 1139 (2013).
3. Kim, K. Y., Cho, D. Y., Cheong, B. K., Kim, D., Horii, H. & Han, S. *J. Appl. Phys.* **113**, 134302 (2013).
4. Bartók, A. P. & Csányi, G. *Int. J. Quant. Chem.* **115**, 1051 (2015).
5. Mocanu, F. C., Konstantinou, K., Lee, T. H., Bernstein, N., Deringer, V. L., Csányi, G. & Elliott, S. R. *J. Phys. Chem. B* **122**, 8998 (2018).
6. Luckas, J., Krebs, D., Salinga, M., Wuttig, M. & Longeaud, C. *Phys. Status Solidi C* **7**, 852 (2010).
7. Rütten, M., Kaes, M., Albert, A., Wuttig, M. & Salinga, M. *Sci. Rep.* **5**, 17362 (2015).
8. Kaes, M. & Salinga, M. *Sci. Rep.* **6**, 31699 (2016).
9. Caravati, S., Bernasconi, M., Kühne, T. D., Krack, M. & Parrinello, M. *Appl. Phys. Lett.* **91**, 171906 (2007).
10. El-Sayed, A. M., Watkins, M. B., Shluger, A. L. & Afanas'ev, V. V. *Microelectronic Engineering* **109**, 68 (2013).
11. El-Sayed, A. M., Watkins, M. B., Afanas'ev, V. V. & Shluger, A. L. *Phys. Rev. B* **89**, 125201 (2014).
12. El-Sayed, A. M., Watkins, M. B., Grasser, T., Afanas'ev, V. V. & Shluger, A. L. *Phys. Rev. Lett.* **114**, 115503 (2015).
13. Konstantinou, K., Duffy, D. M. & Shluger, A. L. *Phys. Rev. B* **94**, 174202 (2016).
14. Dicks, O. A. & Shluger, A. L. *J. Phys.: Condens. Matter* **29**, 314005 (2017).
15. Dicks, O. A., Cottom, J., Shluger, A. L. & Afanas'ev, V. V. *Nanotechnology* **30**, 205201 (2019).
16. Kaviani, M., Strand, J., Afanas'ev, V. V. & Shluger, A. L. *Phys. Rev. B* **94**, 020103 (2016).
17. Strand, J., Kaviani, M. & Shluger, A. L. *Microelectronic Engineering* **178**, 279 (2017).
18. Zipoli, F., Krebs, D. & Curioni, A. *Phys. Rev. B* **93**, 115201 (2016).
19. Branicio, P. S., Bai, K., Ramanarayan, H., Wu, D. T., Sullivan, M. B. & Srolovitz, D. J. *Phys. Rev. Materials* **2**, 043401 (2018).
20. Akola, J. & Jones, R. O. *J. Phys.: Condens. Matter* **20**, 465103 (2008).
21. Caravati, S., Bernasconi, M., Kühne, T. D., Krack, M. & Parrinello, M. *J. Phys.: Condens. Matter* **21**, 255501 (2009).
22. Caravati, S., Bernasconi, M. & Parrinello, M. *Phys. Rev. B* **81**, 014201 (2010).
23. Du, J. & Xiang, Y. *J. Non-Cryst. Solids* **358**, 1059 (2012).

24. Tilocca, A. *J. Chem. Phys.* **139**, 114501 (2013).
25. Prasai, K. & Drabold, D. A. *Nano. Res. Lett.* **9**, 594 (2014).
26. Côté, A. S., Cormack, A. N. & Tilocca A. *J. Mater. Sci.* **52**, 9006 (2017).
27. Flores-Ruiz, H. & Micoulaut, M. *J. Chem. Phys.* **148**, 034502 (2018).
28. Atta-Fynn, R., Drabold, D. A. & Biswas, P. *J. Non-Cryst. Solids: X* **1**, 100004 (2019).
29. Xu, M., Cheng, Y. Q., Sheng, H. W. & Ma, E. *Phys. Rev. Lett.* **103**, 195502 (2009).
30. Mukhopadhyay, S., Sun, J., Subedi, A., Siegrist, T. & Singh, D. J. *Sci. Rep.* **6**, 25981 (2016).
31. Bouzid, A., Ori, G., Boero, M., Lampin, E. & Massobrio, C. *Phys. Rev. B* **96**, 224204 (2017).
32. Perdew, J. P., Ruzsinszky, A., Csonka, G. I., Vydrov, O. A., Scuseria, G. E., Constantin, L. A., Zhou, X. & Burke, K. *Phys. Rev. Lett.* **100**, 136406 (2008).
33. Lee, T. H. & Elliott, S. R. *Adv. Mater.* **29**, 1700814 (2017).
34. Guidon, M., Hutter, J. & VandeVondele, J. *J. Chem. Theory Comput.* **5**, 3010 (2009).
35. Guidon, M., Hutter, J. & VandeVondele, J. *J. Chem. Theory Comput.* **6**, 2348 (2010).
36. Savin, A., Becke, A. D., Flad, J., Nesper, R., Preuss, H. & von Schnering, H. G. *Angew. Chem., Int. Ed.* **30**, 409 (1991).
37. Silvi, B. & Savin, A. *Nature* **371**, 683 (1994).
38. Kalikka, J., Akola, J., Jones, R. O., Kohara, S. & Usuki, T. *J. Phys.: Condens. Matter* **24**, 015802 (2012).
39. Deringer, V. L., Zhang, W., Lumeij, M., Maintz, S., Wuttig, M., Mazzarello, R. & Dronskowski, R. *Angew. Chem., Int. Ed.* **53**, 10817 (2014).
40. Cormack, A. N. & Du, J. *J. Non-Cryst. Solids* **293-295**, 283 (2001).
41. Du, J. & Cormack, A. N. *J. Non-Cryst. Solids* **283**, 69 (2004).
42. Tilocca, A. & de Leeuw, N. H. *J. Mater. Chem.* **16**, 1950 (2006).
43. Pedone, A., Malavasi, G., Menziani, M. C., Cormack, A. N. & Serge, U. *J. Phys. Chem. B* **110**, 11780 (2006).
44. Pedone, A., Malavasi, G., Cormack, A. N., Serge, U. & Menziani, M. C. *Theor. Chem. Acc.* **120**, 557 (2008).
45. Pedone, A., Malavasi, G., Menziani, M. C., Serge, U. & Cormack, A. N. *J. Phys. Chem. C* **112**, 11034 (2008).
46. Xiang, Y., Du, J., Skinner, L. B., Benmore, C. J., Wren, A. W., Boyd, D. J. & Towler, M. R. *RSC Advances* **3**, 5966 (2013).
47. Xiang, Y., Du, J., Smedskjaer, M. M. & Mauro, J. C. *J. Chem. Phys.* **139**, 044507 (2013).
48. Konstantinou, K., Sushko, P. V. & Duffy, D. M. *J. Non-Cryst. Solids* **422**, 57 (2015).
49. Pedesseau, L., Ispas, S. & Kob, W. *Phys. Rev. B* **91**, 134202 (2015).

50. Konstantinou, K., Sushko, P. V. & Duffy, D. M. *Phys. Chem. Chem. Phys.* **18**, 26125 (2016).

REVIEWERS' COMMENTS:

Reviewer #1 (Remarks to the Author):

I do not think the authors answer well all the reviewer's comments. Some new questions have been raised in the manuscript after a large amount of revision.

1 Many AIMD works have been studied the amorphous structures of Ge₂Sb₂Te₅, however, the authors ignored some important references, such as Phys. Rev. Lett. 103, 195502 (2009).

2 During the authors' simulation, the residual stress is as high as 1.7GPa after the geometry optimization. Although the authors claim that this does not affect the quality of the calculation, they should provide other reference to support the claim. Actually, the work PNAS 108, 10410 (2011) has been thoroughly studied the Pressure-induced reversible amorphization and an amorphous–amorphous transition in Ge₂Sb₂Te₅.

3 The conclusions are too long to well organize. Some discussions in conclusions are suggested to move to section results and discussion.

Reviewer #2 (Remarks to the Author):

The authors have done an excellent job in thoroughly addressing the concerns raised by me and the other reviewers. The new version of the manuscript contains additional simulations and analyses that clarify most of the questions I had regarding their study. Hence, I recommend publication of the new version of the paper.

However, I believe that the inability of the GAP potential to yield the "correct" proportion of tetrahedral structures should be briefly mentioned in the paper, since this structural motif may also play a role in the aging process of GST (when I say "correct", I of course mean the proportion obtained by performing GGA simulations, against which the GAP potential was fitted).

Reviewer #3 (Remarks to the Author):

The authors comprehensibly covered all the questions pointed out by the previous review, clarifying all the puzzling points of the original version. In so doing, I now recommend its publication in Nat. Commun. as this work can be now considered an exhaustive, valuable and original study on one of the current uppermost challenge in the field of PCMs materials.

Reply to Reviewer #1

Reviewer #1 Q1

Many AIMD works have been studied the amorphous structures of Ge₂Sb₂Te₅, however, the authors ignored some important references, such as Phys. Rev. Lett. 103, 195502 (2009).

Authors: We do not really understand the comment from the Reviewer, since in our previous response with respect to **Reviewer #3 Q2** we provided a thorough summary about the effects of all the different DFT-schemes on the structure of glassy 225GST that have been reported in the literature, and beyond. We are completely aware of the literature regarding the modelling studies in amorphous phase-change materials, and we would like to clarify that we did not ignore any previous study, even in the first submission. We believe that we made it very clear in our previous response, and also in the manuscript, about how significant is to perform DFT calculations with hybrid functionals with respect to the objective of our work. For the study presented in our manuscript the use of hybrid-DFT calculations is instrumental and of utmost importance in order to estimate correctly the band gap of the glass and establish an accurate description of the electronic structure, which are crucial for the investigation of the mid-gap electronic states. Hybrid-functional DFT calculations were the only choice to identify and characterize the localized defect states in the band gap of glassy 225GST, which then can be ascribed to specific structural features within the amorphous network.

Nevertheless, in the revised manuscript we commented about the studies that report the effects of the different DFT-schemes on the structure of amorphous 225GST (see **page 10 – end of paragraph 2**), while also the PRL reference, mentioned by the Reviewer, already exists in our reference list (**Ref. 43** of the current manuscript). In addition, we have already added two Supplementary Notes, one summarizing the several studies from the literature about this topic (see **Supplementary Note 5**), and another describing why CP2K is an efficient code to perform heavy hybrid-DFT geometry optimizations (see **Supplementary Note 11**).

Therefore, we believe that we have already covered the issue adequately and we do not think that further revisions are needed regarding this aspect.

Reviewer #1 Q2

During the authors' simulation, the residual stress is as high as 1.7GPa after the geometry optimization. Although the authors claim that this does not affect the quality of the calculation, they should provide other reference to support the claim. Actually, the work PNAS 108, 10410 (2011) has been thoroughly studied the Pressure-induced reversible amorphization and an amorphous–amorphous transition in Ge₂Sb₂Te₅.

Authors: We appreciate the comment from the Reviewer and the respective criticism, however we would like to point out the differences between our work and the relevant study from that PNAS paper. For the simulations presented in our work (see also **Supplementary Note 10** for details), we started from a pseudo-random arrangement of the atoms inside a cubic simulation box. Every system was melted at a high temperature, equilibrated to a liquid temperature, and then subsequently quenched to 300K to obtain a glass (and then down to around 0K to perform the electronic-structure calculations). In the PNAS paper, mentioned by the Reviewer, the authors started from a rocksalt-crystalline structure of GST, which was thermalized at 300K with molecular-dynamics simulations, and then the system was subjected to increasing pressures by adjusting the volume of the computational cell. Hence, in the PNAS paper the authors applied pressure in the crystalline phase and then with molecular-dynamics simulations they made their observations. We would like to clarify, that in our modelled systems the residual stress has not been applied externally, but it is due to the NVT molecular-dynamics melt-and-quench simulations followed to generate the amorphous models.

In our previous response we gave a detailed explanation about the fixed-volume approach used in this work to generate the glass models. Matching the simulation cell volume to the target-glass density throughout the melt-and-quench molecular-dynamics simulations corresponds to a robust approach employed and well tested in many prior modelling studies in order to generate glass models representative of the experimental structures (see our reply to **Reviewer #3 Q4**). We fixed the density to the experimental value in order to generate 225GST glass models with good representative atomic structures, which is important with respect to the objective of our study in finding the structural motifs within the amorphous network that are hosting the mid-gap defect electronic states of the glass. We commented in the main manuscript about the resulting total residual stress in the glass models (see **end of page 13**), while a further comment about the fixed-volume approach and the resulting residual stresses in the cell, is given in **Supplementary Note 9**.

Moreover, the differences in the values of the pressure in the computational cells between our glass models and the structures simulated in the PNAS paper are huge. In our modelled systems the residual stress is ~1.5 GPa, whereas in the PNAS paper the authors applied pressures ranging from 10 to 50 GPa (see Fig.4 from the PNAS paper). In addition, the authors

highlighted that the amorphization of the crystalline GST structure is occurring between 18 and 22 GPa of the applied pressures. These values are much larger than the residual stress calculated in our amorphous models, hence we do not believe that similar effects can arise in our simulated systems.

We provided the values for the residual stress in the final glass models, while also we commented about the correlation between these values and the computational methodology employed in our work for the generation of the amorphous models. We believe that the existing revisions in the manuscript and Supplementary Information have covered the issue, therefore we would like not to develop this line of discussion any further at this point.

Reviewer #1 Q3

The conclusions are too long to well organize. Some discussions in conclusions are suggested to move to section results and discussion.

Authors: We agree with the Reviewer and the respective suggestion. Hence, we have worked in the direction to improve the structure of the manuscript and we revised the paper accordingly.

Revisions: We moved the discussions for the spontaneous structural relaxation of the amorphous state from the relevant experimental and modelling studies in our results and discussion section related to the energetic character of the simulated amorphous systems (see **end of page 10 – beginning of page 11**). Our, different, view about the resistance drift, which does not involve structural-ageing effects was placed after the electron-trapping calculations and it was correlated with the respective findings (see **pages 16 and 17**). We finally kept a conclusive paragraph associated with the objective of the study and our simulations (see **page 17**).

Reply to Reviewer #2

Remarks to the Author

The authors have done an excellent job in thoroughly addressing the concerns raised by me and the other reviewers. The new version of the manuscript contains additional simulations and analyses that clarify most of the questions I had regarding their study. Hence, I recommend publication of the new version of the paper. However, I believe that the inability of the GAP potential to yield the "correct" proportion of tetrahedral structures should be briefly mentioned in the paper, since this structural motif may also play a role in the aging process of GST (when I say "correct", I of course mean the proportion obtained by performing GGA simulations, against which the GAP potential was fitted).

Authors: We would really like to thank the Reviewer for appreciating our work related to the additional simulations and the revisions. We would also like to thank the Reviewer, once again, for motivating us to perform some of the extra calculations, like those for the 900-atom glass structure, since they largely increased the scope of our study.

We agree with the Reviewer that the discrepancy between the GAP-MD and the DFT-MD glass structures with respect to the tetrahedral atomic geometries within the amorphous network should be mentioned in the manuscript; therefore, we have already commented in the text about this inability of the potential (see **page 3 – paragraph 2 in the Results and Discussion section**).

Reply to Reviewer #3

Remarks to the Author

The authors comprehensibly covered all the questions pointed out by the previous review, clarifying all the puzzling points of the original version. In so doing, I now recommend its publication in Nat. Commun. As this work can be now considered an exhaustive, valuable and original study on one of the current uppermost challenge in the field of PCMs materials.

Authors: We would really like to thank the Reviewer for appreciating our efforts regarding the revisions, helping us with his/her suggestions to improve the quality of our study, and also for finding our work important with respect to its objective.